# Long ascending propriospinal neurons provide flexible, context-specific control of interlimb coordination

Amanda M Pocratsky[1,2†], Courtney T Shepard[1,2], Johnny R Morehouse[2,3], Darlene A Burke[2,3], Amberley S Riegler[2,3], Josiah T Hardin[4], Jason E Beare[2,5], Casey Hainline[4], Gregory JR States[1,2], Brandon L Brown[2], Scott R Whittemore[1,2,3], David SK Magnuson[1,2,3,4]*

[1]Department of Anatomical Sciences and Neurobiology, University of Louisville, Louisville, United States; [2]Kentucky Spinal Cord Injury Research Center, University of Louisville, Louisville, United States; [3]Department of Neurological Surgery, University of Louisville, Louisville, United States; [4]Speed School of Engineering, University of Louisville, Louisville, United States; [5]Cardiovascular Innovation Institute, Department of Physiology and Biophysics, University of Louisville, Louisville, United States

**\*For correspondence:**
dsmagn01@louisville.edu

**Present address:** †Department of Neuromuscular Diseases, Institute of Neurology, University College London, London, United Kingdom

**Competing interests:** The authors declare that no competing interests exist.

**Abstract** Within the cervical and lumbar spinal enlargements, central pattern generator (CPG) circuitry produces the rhythmic output necessary for limb coordination during locomotion. Long propriospinal neurons that inter-connect these CPGs are thought to secure hindlimb-forelimb coordination, ensuring that diagonal limb pairs move synchronously while the ipsilateral limb pairs move out-of-phase during stepping. Here, we show that silencing long ascending propriospinal neurons (LAPNs) that inter-connect the lumbar and cervical CPGs disrupts left-right limb coupling of each limb pair in the adult rat during overground locomotion on a high-friction surface. These perturbations occurred independent of the locomotor rhythm, intralimb coordination, and speed-dependent (or any other) principal features of locomotion. Strikingly, the functional consequences of silencing LAPNs are highly context-dependent; the phenotype was not expressed during swimming, treadmill stepping, exploratory locomotion, or walking on an uncoated, slick surface. These data reveal surprising flexibility and context-dependence in the control of interlimb coordination during locomotion.

## Introduction

Locomotion is a fundamental behavior that allows animals to move through the environment to forage, escape predators, or simply explore. Its expression is initiated supraspinally by various brain nuclei that provide locomotor command cues to spinal circuits, the downstream effectors of movement (*Caggiano et al., 2018*). Ultimately, it is the responsibility of the spinal cord circuitry to organize limb movements into the stepping patterns that are defined as locomotor gaits (*Orlovskiĭ et al., 1999*).

The two enlargements of the spinal cord serve as primary sites for the organization of forelimb and hindlimb movements, respectively (*Cazalets et al., 1995*; *Grillner, 1981*). Embedded within each enlargement are limb-specific central pattern generators (CPGs), each tasked with generating the respective patterns of limb movement (*Kiehn, 2006*). Through a distributed network of intra- and inter-enlargement connections, the fore- and hindlimb CPGs orchestrate the rhythm and pattern features of locomotion, including those associated with speed-dependent gaits: walk-trot, gallop, and bound (*Brockett et al., 2013*; *Miller and van der Meché, 1976*; *Miller and Van der Burg,*

*1973*; *Miller et al., 1975*; *Juvin et al., 2005*; *Juvin et al., 2007*). Two classes of inter-enlargement spinal neurons are thought to coordinate forelimb-hindlimb movements: long ascending propriospinal neurons (LAPNs) and long descending propriospinal neurons (LDPNs) (*Miller and van der Meché, 1976*; *Miller et al., 1975*; *Juvin et al., 2005*).

LDPNs reside in the cervical enlargement and project broadly to multiple sites throughout the spinal cord, including the lumbar enlargement (*Reed et al., 2006*; *Ni et al., 2014*; *Alstermark et al., 1987*; *Giovanelli Barilari and Kuypers, 1969*). Electrophysiological studies in the cat suggest that LDPNs are primarily involved in postural control by way of relaying proprioceptive inputs from the head and neck to the hindlimb motor pools (*Alstermark et al., 1987*). Using mouse genetics and viral technology, Ruder and colleagues revealed that not only do LDPNs ensure postural stability, but they also secure interlimb coordination during high-speed locomotion (*Ruder et al., 2016*).

Considerably less is known about the "reciprocal" inter-enlargement pathway: the LAPNs. Studies performed in the cat, rat, and mouse collectively reveal that LAPNs are a heterogeneous network of both ipsi- and contralaterally projecting neurons with mixed neurotransmitter phenotypes (excitatory and inhibitory) (*Reed et al., 2006*; *Giovanelli Barilari and Kuypers, 1969*; *Ruder et al., 2016*). The functional role of LAPNs in vivo remains unknown. Here, we used reversible synaptic silencing of the LAPNs to determine their role during locomotion. Our data suggest that LAPNs form a flexible, task-specific network for securing interlimb coordination of each limb pair (at the forelimb and hindlimb girdles, respectively) in a highly context-driven manner.

## Results

### Histological detection of conditionally silenced LAPNs

Spinal circuits located in the intermediate gray matter of the caudal cervical and rostral lumbar segments are the primary rhythmogenic sites for locomotor output (*Cazalets et al., 1995*; *Juvin et al., 2005*; *Ballion et al., 2001*). LAPNs, which are primarily embedded within the intermediate gray matter of the rostral lumbar segments, send ipsilateral or contralateral projections to the caudal cervical enlargement with sparse resident projections within the lumbar neuraxis (*Figure 1—figure supplement 1*). Given the critical involvement of cervical and lumbar CPGs for locomotion and the anatomical profile of the long ascending projections which connect these rhythmogenic foci, we set out to silence LAPNs in the freely behaving adult rat. We used the dual-virus TetOn system originally developed by Isa and colleagues (18), which allows doxycycline-induced reversible silencing of anatomically defined projection neurons (details in methods). Using two pairs of microinjections into the intermediate gray matter, we simultaneously targeted ipsilateral and commissural LAPNs that connect the key rhythmogenic foci (L1-L3 and C6-C8) reasoning that their silencing would lead to overt changes in hindlimb-forelimb coordination (*Figure 1a*). Behavioral testing was performed at Baseline (prior to injection), pre-silencing (Pre-Dox1), during Dox$^{On}$ conditional silencing of LAPNs, and post-silencing (Dox$^{Off}$) (*Figure 1b*). Repeat assessments were performed one month later (Dox2).

To confirm that any silencing-induced behavioral changes were concomitant with eTeNT-expression in LAPNs, animals were euthanized during Dox2$^{On}$ LAPN silencing, following terminal behavioral assessments, and the spinal cords were processed for eTeNT.EGFP immunoreactivity. Histological analyses of the caudal cervical enlargement revealed that eTeNT.EGFP-expressing putative fibers surrounded and closely apposed neuronal somata and processes (*Figure 1c–f*). Moreover, eTeNT.EGFP co-localized with the synapse-related markers synaptophysin (*Figure 1g–h*), vesicular glutamate transporter 2 (*Figure 1i*, excitatory neurotransmitter), and vesicular GABA transporter (*Figure 1j*, inhibitory neurotransmitter). Collectively, these data suggest that the cervical projections derived from double-infected LAPNs express eTeNT and were silenced in vivo.

We next screened for the double-infected LAPN somata in the lumbar spinal cord. Using immunoperoxidase to enhance the eTeNT.EGFP signal, we observed EGFP+ neurons distributed throughout the rostral lumbar enlargement (*Figure 1l–n*, filled arrowheads). Intermingled with the double-infected LAPNs were non-infected lumbar neurons (open arrowheads). Isotype controls revealed little-to-no immunoreactivity suggesting that the histological detection of the conditionally expressed eTeNT.EGFP was specific (*Figure 1k,o–p*).

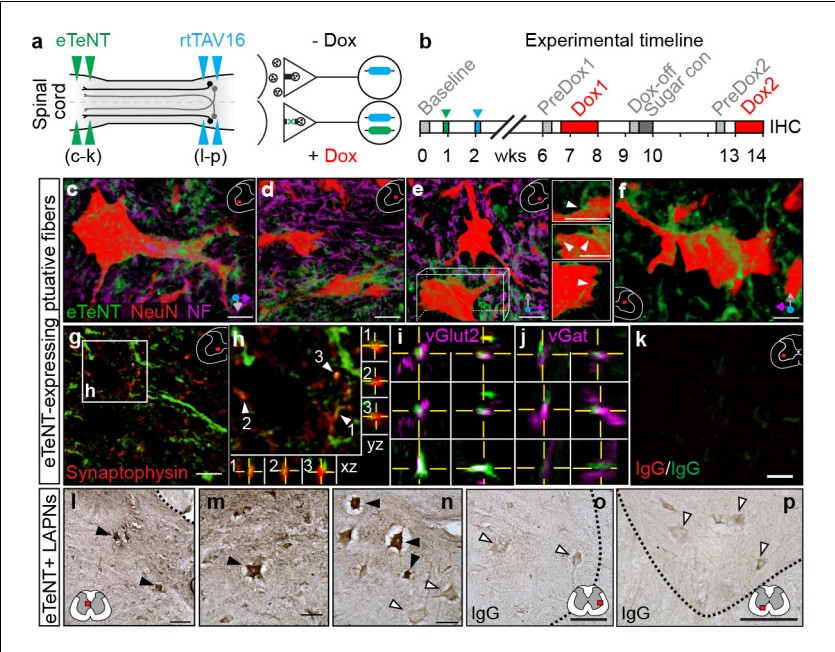

**Figure 1.** Histological detection of putatively silenced long ascending propriospinal neurons (LAPNs). (a–b) Experimental design (see Materials and methods for details). (c–f) Volume rendered, high magnification images showing enhanced eTeNT.EGFP putative fibers (green) surrounding NeuN-stained neurons (red) and neurofilament-marked neural processes (magenta) in the cervical spinal cord (100x; x-y-z axis orientation shown in bottom right). Neuron in panel e is rotated about x-y-z axis to show eTeNT.EGFP fibers surrounding somata (inset panels right side). Neurofilament staining excluded in panel f for clarity (eTeNT.EGFP enshrouding cervical neuron). (g–h) eTeNT.EGFP signal co-localizes with synaptophysin (red). XZ-YZ orthogonal cross-sections through putative synapses shown in panel (h). eTeNT.EGFP signal co-localizes with the excitatory neurotransmitter marker vesicular glutamate transporter 2 (i, vGlut2; magenta) as well as the inhibitory neurotransmitter marker vesicular GABA transporter (j, vGat; magenta) (XZ-YZ orthogonal cross-sections shown). (k) Isotype controls revealed little-to-no immunoreactivity (IgG controls for synaptophysin and eTeNT.EGFP shown). (l–n) DAB enhancement of eTeNT.EGFP at the lumbar segments revealed dark immunoreactive neurons in the rostral lumbar segments (filled arrowheads) intermingled with DAB-negative neurons (open arrowheads). (o–p) Isotype control revealed little-to-no immunoreactivity. (c–k Scale bar = 25 µm; g,k scale bar = 10 µm; l,o,p Scale bar = 100 µm; m,n Scale bar = 50 µm).

The online version of this article includes the following source data and figure supplement(s) for figure 1:

**Figure supplement 1.** Long ascending propriospinal neurons (LAPNs) are a bilaterally distributed pathway throughout the rostral lumbar enlargement with modest local projections.

**Figure supplement 1—source data 1.** contains the source data for CTB labeled cell body counts.

## LAPNs organize interlimb coupling at each girdle during overground stepping

After validating that double-infected LAPNs conditionally expressed eTENT.EGFP in the presence of doxycycline, we next set out to determine the functional consequences of silencing this inter-enlargement pathway in the freely behaving adult rat.

Prior to silencing, animals stepped in a stereotypic walk or trot-like gait with the left-right limbs moving out-of-phase (alternating) at each girdle and the contralateral hindlimb-forelimb pairs moving in-phase (synchronously) (*Figure 2a–c*). Conditionally silencing LAPNs resulted in a striking spectrum of stepping behaviors, ranging from mild disruptions in left-right hindlimb alternation to a half-bound-like gait where the hindlimbs moved synchronously as the forelimbs "galloped," all the way to a full-bound where both the left-right forelimbs and hindlimbs moved synchronously (*Video 1*). The stepping behavior reverted back to the usual walk and trot-like gaits when silencing was reversed by removing Dox (*Figure 2—figure supplement 1*). Re-silencing LAPNs one month later reproduced and, in some cases, even enhanced these effects (*Figure 2—figure supplement 1e*).

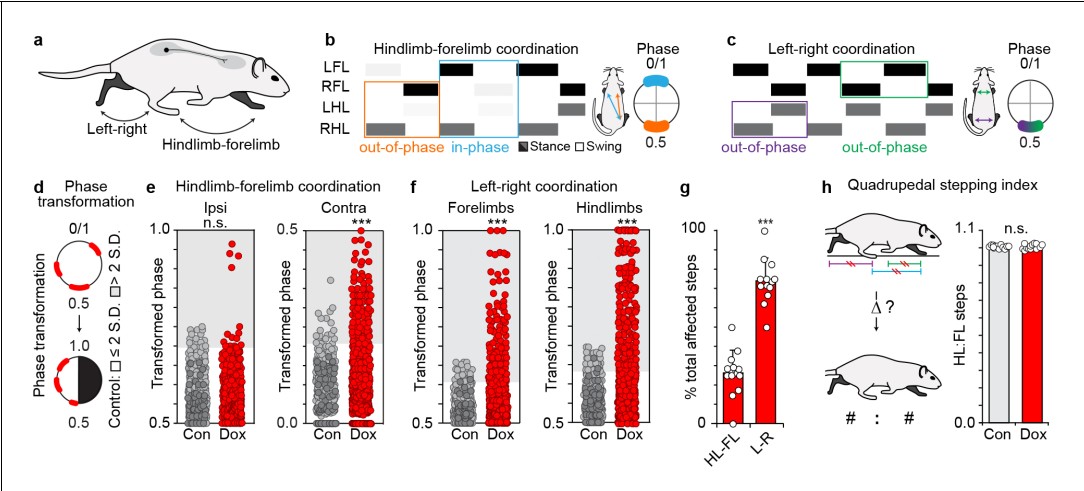

**Figure 2.** Silencing long ascending propriospinal neurons (LAPNs) disrupts intra-girdle movements during overground stepping. (**a–c**) Representative swing-stance graphs of stepping behaviors observed at control time points. Left: orange = homolateral HL-FL movements (out-of-phase, 0.5), blue = diagonal HL-FL movements (in-phase, 0.0/1.0). Right: green = left -right forelimb, purple = left-right hindlimb, each out-of-phase. Insets = one complete stride cycle (right limb reference). (**d**) Circular 0–1 phase data are transformed into a linear scale (0.5–1.0 or 0.0–0.5). (**e**) Left: silencing LAPNs does not disrupt homolateral ("ipsi") HL-FL coordination (# steps beyond control variability: Control n = 19/480 [3.95%] vs Dox n = 17/600 [2.83%]; p=0.31, z = 1.01, Binomial Proportion Test; circles = individual step cycles; shaded region = values beyond control variability). Right: diagonal ("contra") HL-FL coordination is significantly disrupted (Control n = 17/480 [3.54%] vs Dox n = 98/600 [16.33%]; ***p<0.001, z = 7.47). (**f**) Silencing LAPNs significantly disrupts left-right forelimb and left-right hindlimb coordination, respectively (forelimbs: Control n = 26/480 [5.42%] vs Dox n = 135/600 [22.50%]; p<0.001, z = 8.57; hindlimbs: Control n = 26/480 [5.42%] vs Dox n = 177/600 [29.50%]; ***p<0.001, z = 11.31). (**g**) Silencing LAPNs disrupts left-right movements more than hindlimb-forelimb (% total altered steps: hindlimb-forelimb 26.20 ± 3.02% vs left-right 73.80 ± 3.37%; ***p<0.001, critical t = 2.17, paired t-test; bars = group mean± S.D.; circles=% total steps taken that are altered for individual animals). (**h**) The quadrupedal stepping index remained unchanged during silencing (Control: 100.78 ± 0.87 vs Dox: 100.76 ± 1.55; p=0.97, critical t = 2.17; paired t-test).

The online version of this article includes the following source data and figure supplement(s) for figure 2:

**Source data 1.** Contains the source data for step ratio measures.

**Source data 2.** Contains the source data for the magnitude of change of step ratio measures.

**Source data 3.** Contains the source data for the interlimb coordination measures.

**Figure supplement 1.** Silencing long ascending propriospinal neurons (LAPNs) disrupts interlimb coordination during overground locomotion.

These data suggest that LAPNs secure multiple interlimb coupling patterns, not strictly hindlimb-forelimb coordination as we initially hypothesized.

In light of the unexpected changes to overall stepping behavior, we quantified the silencing-induced disruption of interlimb coordination. We first linearized the circular phase data to account for inter-animal variability in preferred lead limb during stepping (*Pocratsky et al., 2017*; *Figure 2d*) (e.g. for the left-right hindlimbs, coordination values of 0.25 or 0.75 are both gallop patterns). We then pooled the phase data from all control time points, calculated the mean temporal relationship for each limb pair, and set a control threshold based on normal variability observed during overground stepping (see methods for details) (*Pocratsky et al., 2017*).

When we gated our analyses to hindlimb-forelimb coordination, we observed an interesting dichotomy in the functional consequences of silencing LAPNs. Contralateral hindlimb-forelimb coordination was selectively disrupted with a significant increase in the proportion of steps that deviated beyond control variability (*Figure 2e*, right panel; *Figure 2—figure supplement 1a–b*). Coefficient of variation analyses substantiated this outcome, revealing an overall increase in the variability observed in hindlimb-forelimb coordination during silencing (CoV; Con vs Dox, 6.86 ± 1.17 vs 9.86 ± 3.91; p<0.05, paired t-test). Conversely, ipsilateral hindlimb-forelimb coordination remained intact (*Figure 2e*, left panel). Switching focus to intra-girdle movements revealed an even more intriguing result. Silencing LAPNs profoundly affected left-right coordination at each girdle (*Figure 2f*, *Figure 2—figure supplement 1c–d*) such that their functional decoupling allowed the full range of

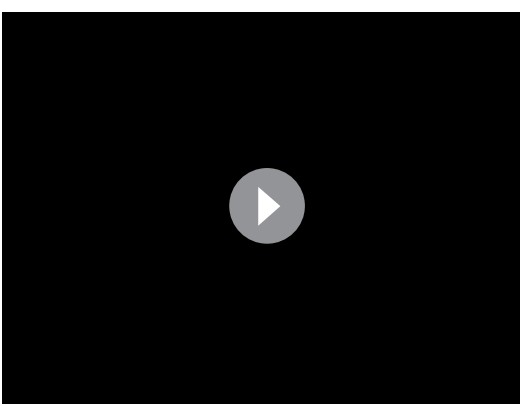

**Video 1.** Conditionally silencing long ascending propriospinal neurons (LAPNs) disrupts interlimb coordination during overground stepping. Dox<sup>On</sup> videos shown from two independent experiments, three separate animals at Dox1<sup>On</sup> Day 8 of LAPN silencing. Videos shown from the same animal at 1x, 0.5x, and 0.25x speed.
https://elifesciences.org/articles/53565#video1

possible stepping phases to be expressed (*Supplementary file 1*; forelimb CoV: Con vs Dox, 9.81 ± 1.24 vs 18.08 ± 7.94, p<0.005; hindlimb CoV: 12.10 ± 2.20 vs 26.38 ± 14.39; p<0.005).

We then pooled the stepping bouts with altered coordination and compared the frequency of perturbed patterns (hindlimb-forelimb vs intra-girdle left-right). We found that perturbations to left-right alternation at each girdle was the primary deficit during LAPN silencing (*Figure 2g*). Moreover, when we screened for concurrent changes across the limb pairs, we found that the majority of hindlimb-forelimb perturbations were concomitant with intra-girdle left-right disruptions, but not vice versa (*Figure 2—figure supplement 1f*). These data suggest that LAPNs play a key role in securing left-right coordination at each girdle, and that changes to inter-girdle (hindlimb-forelimb) coordination are likely indirect. Despite the silencing-induced freedom in pattern expression observed within each girdle, all four limbs continued to step in a fixed 1:1 ratio (*Figure 2h*), indicating that other key features of locomotor control remain intact.

## Intralimb coordination and postural control endure despite silencing-induced interlimb discoordination

During stepping, temporal information is distributed between (interlimb) and within (intralimb) each limb (*Kiehn, 2006*). Given the overt disruption to interlimb coordination, we set out to determine if intralimb movements were also affected during LAPN silencing. Using a three-segment, two-angle model of the hindlimb (*Pocratsky et al., 2017*; *Kuerzi et al., 2010*), we quantified both the spatial and temporal properties of intralimb coordination during stepping (*Figure 3a*).

At control time points, the hindlimbs showed normal range-of-motion throughout the step cycle (*Figure 3b*) and normal proximal-to-distal coordination (*Figure 3c*). This spatial coordination persisted during silencing, even during bouts of synchronous stepping events. We next examined the temporal features of intralimb movement. Typically, peak excursion of both the proximal and distal limb components occurs at the end of stance phase just prior to lift-off (*Figure 3d*; *Pocratsky et al., 2017*). This salient feature of intralimb coordination also remained intact during LAPN silencing (*Figure 3e*), indicating that altered coordination between limb pairs did not affect the coordination of the limb itself (*Figure 3f*).

Given the generalized disruption to interlimb coordination, we also explored how balance/postural stability is affected during LAPN silencing. LDPNs, the pathway reciprocal to LAPNs, play a key role in this supportive feature of locomotion (*Ruder et al., 2016*). To interrogate postural stability, animals were challenged using a series of graded tasks with increased demand for balance control. Posturally-challenged animals often externally rotate their hindpaws during stepping to increase the overall base-of-support (*Basso et al., 1995*). We found no increase in the per-step angular rotation of the hindpaws during LAPN silencing, suggesting that base-of-support remained unchanged despite the disrupted phase relationship between limb pairs at each girdle (*Figure 3g*). Similarly, silencing LAPNs did not lead to increased footfalls on the narrow beam or horizontal ladder (*Figure 3h*), tasks with increased demand for balance control. Silencing LAPNs also did not negatively impact the frequency and duration of spontaneous rearing events, a task where quadrupedal animals stand bipedally (*Figure 3i*). Finally, animals were challenged with lap swimming, a task where the limbs are unloaded and postural control is essential for effective hindlimb-driven propulsion (*Gruner and Altman, 1980*). Using the body angle relative to the water surface as a proxy for trunk stability, we again found that LAPN silencing did not affect overall postural control (*Figure 3j*). Thus,

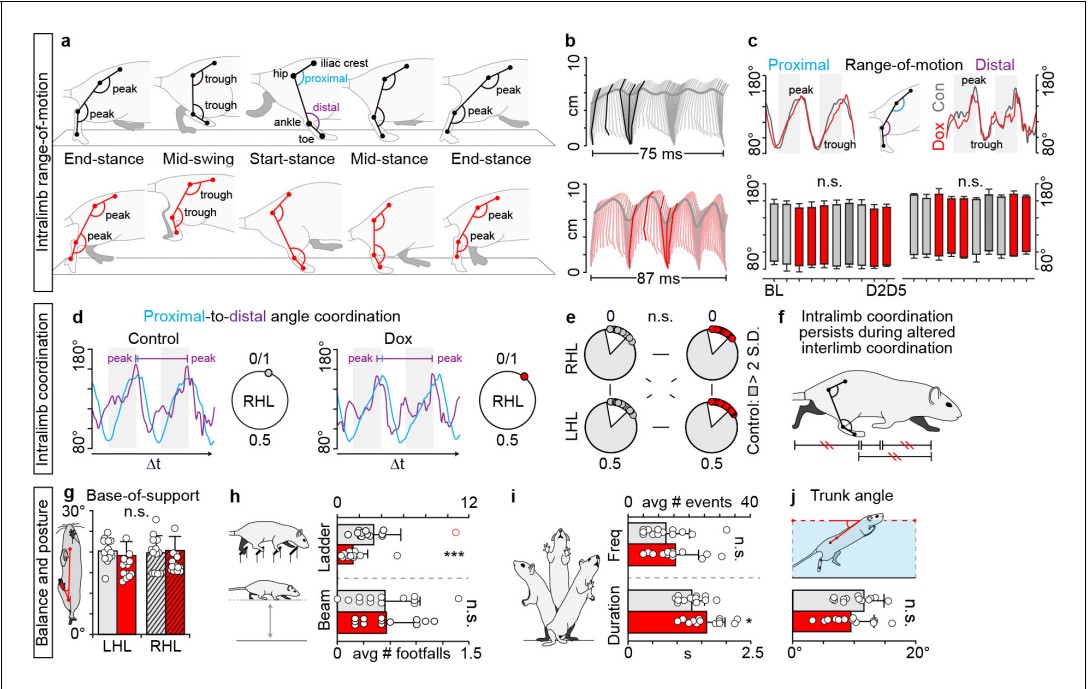

**Figure 3.** Intralimb coordination and postural control endures despite silencing-induced generalized interlimb discoordination. (**a**) Three-segment (iliac crest-hip, hip-ankle, ankle-toe), two-angle model of intralimb coordination. Five phases of step cycle illustrated with corresponding hindlimb range-of-motion (peak-to-trough excursion of the proximal and distal angles) and intralimb kinematics (**b**). (**c**) Range-of-motion was not altered during silencing (right hindlimb shown, group average ± S.D. [Baseline to Dox2On-D5]; p>0.5, mixed model ANOVA, Bonferroni *post hoc*). (**d**) Representative example of proximal-to-distal temporal coordination for one stride cycle (temporal overlap in peak angular excursions). Intralimb coordination plotted on circular graph where 0 denotes in-phase coordination. (**e**) Silencing long ascending propriospinal neurons (LAPNs) did not disrupt the proximal-to-distal temporal relationship across the hindlimb segments (p>0.5 for all comparisons; Watson's U (*Orlovskiï et al., 1999*) test). White inset = control variability. Individual circles = peak to-peak proximal or distal excursion for one stride cycle. (**f**) Summary schematic. (**g**) Silencing LAPNs did not affect hindlimb base-of-support during overground stepping (Baseline vs Dox1On-D5: left hindlimb, 20.23 ± 3.00˚ [n = 220 steps] vs 19.13 ± 3.38˚ [n = 223 steps], p=0.31; right hindlimb, 19.76 ± 4.19˚ [n = 227 steps] vs 20.37 ± 3.39˚ [n = 229 steps], p=0.62; paired t-tests). (**h**) The number of hindlimb foot falls on the ladder significantly decreased during silencing versus control (Control 3.33 ± 2.47 vs DoxOn1.43 ± 1.33, **p<0.01; excluding outlier [red circle] yielded similar results – see Materials and mmethods for details). No significant differences were detected on the beam (Control 0.55 ± 0.38 vs Dox 0.55 ± 0.32, p=0.96). (**i**) Frequency of spontaneously evoked rearing events remained unchanged during silencing (Control 7.62 ± 4.89 vs Dox 9.46 ± 4.74, p=0.29). There was a slight, but significant increase in the duration of the rearing events during silencing (Control 1.56 ± 0.31 s vs Dox 1.92 ± 0.47 s, p=0.045). (**j**) Trunk angle during swimming remained unchanged during silencing (Control 10.23 ± 2.87˚ vs Dox 9.49 ± 3.78˚, p=0.54). Data shown from N = 13 animals. Circles = individual averages; bars = group average± S.D.

The online version of this article includes the following source data for figure 3:

**Source data 1.** Contains the source data for trunk angle measures.
**Source data 2.** Contains the source data for intralimb range-of-motion.
**Source data 3.** Contains the source data for intralimb coordination measures.
**Source data 4.** Contains the source data for foot faults on the narrow beam.
**Source data 5.** Contains the source data for hindlimb base-of-support.

silencing LAPNs leads to a generalized disruption of interlimb coordination without altering intralimb coordination or overall balance/postural control, key features that are required for effective locomotion.

## Silencing LAPNs disrupts interlimb coordination independent of other salient features of locomotion

Hallmark features of locomotion are speed-dependent changes in interlimb coordination that are classified into different locomotor gaits, each defined by a unique set of limb coupling patterns (*Hildebrand, 1965*). As each gait is expressed as a function of speed, the underlying spatiotemporal features of limb movement predictably change (*Lemieux et al., 2016*). This fundamental relationship

is highlighted in data collected from age-matched control rats assessed in a three meter long runway that allowed the full range of speed-dependent gaits to be expressed (*Figure 4a*, *Figure 4—figure supplement 1*; see methods for detail). As the speed increased (with concomitant changes in

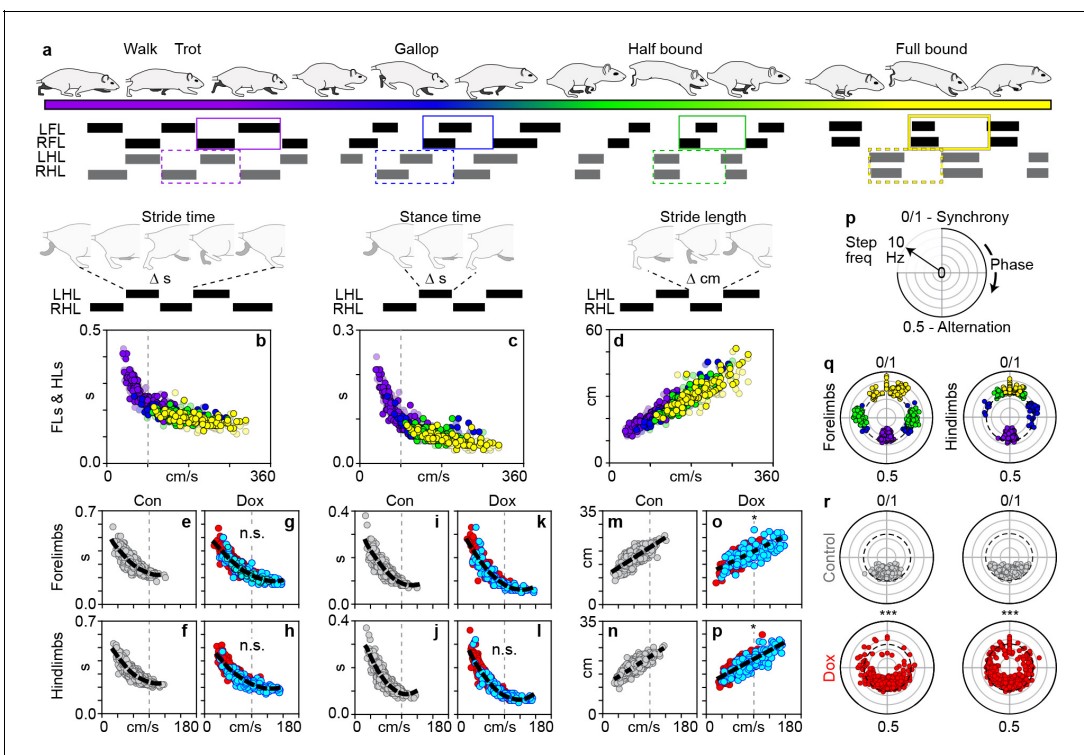

**Figure 4.** Silencing long ascending propriospinal neurons (LAPNs) disrupts interlimb coordination independent from the salient features of locomotion. (a) Schematic illustrating the speed-dependent gaits with representative swing-stance graphs (purple = walk trot; blue = gallop; green = half-bound; yellow = full-bound). (b) Schematic illustrating stride time (duration of one stride) and its normal relationship with speed for the volitionally expressed gaits (N = 12 age-matched controls, see methods; circles = individual steps). This relationship persisted during silencing (forelimbs: c,d, Control: p<0.001, $r_S$ = −0.845, $R^2$ = 0.714 [n = 480 steps] vs Dox: p<0.001, $r_S$ = −0.889, $R^2$ = 0.790 [n = 600]) (hindlimbs: e,f, Control: p<0.001, $r_S$ = −0.864, $R^2$ = 0.746 vs Dox: p<0.001, $r_S$ = −0.908, $R^2$ = 0.824; steps with silencing-induced altered coordination shown in blue for clarity; altered step defined as step cycle with a phase relationship that deviates beyond control variability; dashed line = line of best fit). (g) Schematic illustrating stance time (duration of paw contact for one stride) and its normal relationship with speed. This relationship persisted during silencing (h,i, forelimbs, Control: p<0.001, $r_S$ = −0.905, $R^2$ = 0.819 vs Dox: p<0.001, $r_S$ = −0.929, $R^2$ = 0.863) (j,k, hindlimbs, Control: p<0.001, $r_S$ = −0.901, $R^2$ = 0.812 vs Dox: p<0.001, $r_S$ = −0.946, $R^2$ = 0.895). (l) Schematic illustrating stride length (distance traveled for one stride) and its normal relationship with speed. This relationship persisted during silencing (m,n, forelimbs, Control: p<0.001, $r_S$ = 0.784, $R^2$ = 0.615 vs Dox: p<0.001, $r_S$ = 0.736, $R^2$ = 0.582; o,p, hindlimbs, Control: p<0.001, $r_S$ = 0.801, $R^2$ = 0.642 vs Dox: p<0.001, $r_S$ = 0.787, $R^2$ = 0.619). There was a slight change in the slopes for the lines of best fit for stride length versus speed during silencing (n, *p<0.05; t = 2.18; p, *p<0.05, t = 2.42). (q) Phase-frequency plot illustrating phase change as a function of frequency. (r) Left-right forelimb and left-right hindlimb phase-frequency relationships for the speed-dependent gaits (dashed circle = 5 Hz transition zone from the walk-trot to gallop *Gillis and Biewener, 2001*; *Muir and Whishaw, 2000*). Silencing LAPNs functionally decoupled the left-right fore- and hindlimbs, respectively (s, left: forelimbs, Control vs Dox, ***p<0.001, $U^2$ = 0.67, $n_1$ = 123, $n_2$ = 187, Watson's U (*Orlovskiĭ et al., 1999*) test; (s), right: hindlimbs: Control vs Dox, ***p<0.001, $U^2$ = 1.45, $n_1$ = 131, $n_2$ = 204; refer to *Supplementary file 2*; white inset denotes control variability, circles denote individual step cycles). The decoupled limb pairs stepped at Control-level frequencies (forelimbs, Control: 99.80% [n = 479/480] at ≤5 Hz; Dox: 92.50% [n = 555/600]; hindlimbs, Control: 100% [n = 480/480] at ≤5 Hz; Dox: 95.30% [n = 572/600]).

The online version of this article includes the following source data and figure supplement(s) for figure 4:

**Source data 1.** Contains the source data for stereotypical gait measures.
**Source data 2.** Contains the source data for phase/frequency relationship.
**Source data 3.** Contains the source data for spatiotemporal relationship.
**Figure supplement 1.** Stereotypic limb coupling patterns for speed-dependent, volitionally expressed locomotor gaits.
**Figure supplement 2.** Salient features of locomotion remain unaffected during silencing-induced disruption to interlimb coordination.
**Figure supplement 2—source data 1.** Contains the source data for FL-HL spatiotemporal measures.
**Figure supplement 3.** Rhythmic locomotor output within and between limb girdles remains coupled during long ascending propriospinal neuron (LAPN) silencing.

interlimb coordination and gait), the stride and stance durations decreased while the stride lengthened (*Figure 4b–d*).

Strikingly, this fundamental feature of locomotor control was unaffected during LAPN silencing. Despite the altered temporal coupling patterns expressed at the forelimbs, hindlimbs, and hindlimb-forelimb limb pairs, the spatiotemporal relationships of limb movements and speed remained intact (*Figure 4e–p*; blue circles = altered step cycles; *Figure 4—figure supplement 2a–l*). We saw no changes to the overall stride, stance, and swing durations (*Figure 4—figure supplement 2m*). Individual time point comparisons substantiated these results (*Supplementary file 1*).

Given the saliency of the intact locomotor features in the face of overt changes to interlimb coordination, we next explored the underlying stepping rhythm. We first examined the phase-frequency relationship for the left-right, fore- and hindlimb pairs. We plotted the left-right coordination value of each step taken (*Figure 4q*, ranging from 0 to 1) relative to the underlying step frequency with which it occurred (concentric circles of increasing frequency). The typical phase-frequency relationship is highlighted in our volitional gait dataset from age-matched control animals. Left-right alternation typically occurs at lower step frequencies (indicative of a walk-trot gait) (*Figure 4r*, purple circles). At higher step frequencies, the left-right limb pairs adopt a phase-shifted expression pattern (indicative of a gallop, green). At even greater step frequencies, the half or full-bound emerges wherein the hindlimbs move synchronously while the forelimbs adopt an asynchronous (half-bound, green) or synchronous-like stepping pattern (full-bound, yellow).

At control time points, the left-right limb pairs at each girdle primarily alternated with the majority of steps remaining below a 5 Hz step frequency (*Figure 4r*, top panels). Silencing LAPNs functionally decoupled the left-right limb pairs at each girdle, as revealed by the phasic dispersion throughout the polar plot (*Figure 4s*, bottom panels). Similar results were found following time point comparisons as well as parametric analyses on related measures including phasic concentration and circular variance (*Supplementary file 2*).

Despite the temporal decoupling of the fore- and hindlimb pairs, stepping frequencies remained similar to those of control time points ($\leq$5 Hz). This led us to further explore the underlying stride duration within and between the girdles, with and without controlling for the effect of speed. Once again, silencing LAPNs had no impact on the underlying locomotor rhythm (*Figure 4—figure supplement 3a–d*). As a more sensitive assessment, we compared the stride durations between various limb pairs on a moment-by-moment basis. Each limb pair maintained a predictable relationship in the per-step stride duration despite the silencing-induced disruption to left-right coordination at each girdle (*Figure 4—figure supplement 3e–h*). Together, these data suggest that the rhythm of locomotor output is maintained despite the silencing-induced decoupling of limb pairs, indicating that temporal coordination can be selectively manipulated in an otherwise precisely controlled system.

## Silencing-induced disruption to interlimb coordination is context-dependent

Thus far, results suggest that LAPNs coordinate interlimb movement during volitional overground stepping. To generalize the functional importance of LAPNs beyond this select condition, we assessed interlimb coordination across various locomotor tasks, behavioral modes, and external environments.

We first queried a different locomotor task: treadmill-based stepping. We found that intra-girdle left-right alternation was preferentially affected during overground locomotion as compared to treadmill stepping (*Figure 5a*; *Video 2*; *Figure 5—figure supplement 1d–f*; *Supplementary file 3*).

We then examined interlimb coordination during exploratory-like versus non-exploratory-like locomotion. Exploratory-like stepping was defined as overground locomotor passes where the snout was pointed down and was in close proximity to the ground (see methods for details) (*Video 3*). The non-exploratory stepping mode is the curated dataset shown thus far (*Figure 5b*, right panel, included for comparison purposes). In contrast to non-exploratory locomotion (snout up, "going from A to B"), silencing LAPNs had little-to-no effect on interlimb coordination during exploratory-like locomotion (*Figure 5b*, left panel; *Video 3*; *Figure 5—figure supplement 1g–i*; *Supplementary file 3*).

In a separate experiment, animals were tested on two stepping surfaces with different coefficients of friction: an uncoated acrylic surface (CoF: 0.44) and a Sylgard-coated acrylic surface (CoF: 1.73).

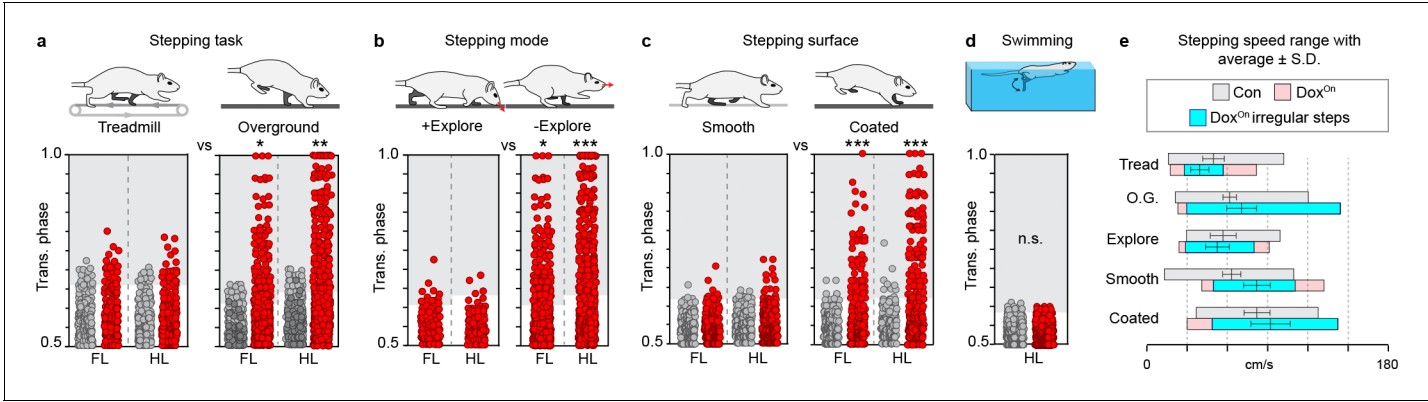

**Figure 5.** Silencing-induced disruption to interlimb coordination occurs in a task-specific, context-driven manner. (a) Intra-girdle left-right coordination was affected to a greater extent during overground stepping as compared to treadmill during long ascending propriospinal neuron (LAPN) silencing (forelimbs, overground n = 135/600 [22.50%] vs treadmill n = 22/151 [17.05%], *p<0.05 [z = 2.38]; hindlimbs, overground n = 177/600 [29.50%] vs treadmill n = 28/151 [22.76%], **p<0.01 [z = 2.99]; *Figure 5—figure supplement 1a–f; Supplementary file 3*). (b) Silencing LAPNs does not affect interlimb coordination during exploratory-like stepping as compared to a more "directed" stepping mode ("going from A to B") (Dox$^{On}$ forelimbs, non-exploratory overground n = 135/600 [22.50%] vs exploratory overground n = 13/95 [13.68%], *p<0.05 [z = 2.25]; Dox$^{On}$ hindlimbs, non-exploratory overground n = 177/600 [29.50%] vs exploratory overground n = 7/95 [7.37%], **p<0.001 [z = 6.78]; *Figure 5—figure supplement 1g–i; Supplementary file 3*) (see Materials and methods for details). Silencing LAPNs does not affect intra-girdle left-right coordination while stepping on an uncoated plexiglass surface as compared to a Sylgard-coated base (N = 8 animals from a separate set of experiments; see methods for details; forelimbs, uncoated plexiglass n = 11/166 [6.63%] vs Sylgard-coated n = 39/170 [22.94%], ***p<0.001 [z = 4.34]; hindlimbs, uncoated plexiglass n = 12/166 [7.23%] vs Sylgard-coated n = 60/170 [35.29%], ***p<0.001 [z = 2.99]; *Figure 5—figure supplement 1j–o; Supplementary file 3*). (d) Silencing LAPNs did not affect left-right hindlimb alternation during swimming (n = 2/390 and 0/390 stroke cycles at Control and Dox, respectively; deviated beyond control variability; p>0.5 [z = 1.0]; *Figure 5—figure supplement 1p; Supplementary file 3*). Data shown in a,b, and d are from N = 13 animals. Data shown in c are from separate set of N = 8 animals. Circles = individual step or stroke cycles. Shaded region denotes variability beyond that observed at control time points for each condition described.

The online version of this article includes the following source data and figure supplement(s) for figure 5:

**Source data 1.** Contains the source data for phase during exploratory walking.
**Source data 2.** Contains the source data for phase during treadmill walking.
**Source data 3.** Contains the source data for phase on different surfaces.
**Source data 4.** Contains the source data for phase during swimming.
**Figure supplement 1.** Interlimb coordination is disrupted in a context-driven manner.
**Figure supplement 2.** Silencing-induced changes to overground stepping occurred at a speed range which is shared across all behavioral contexts.

Silencing LAPNs significantly affected left-right alternation when animals stepped on the Sylgard coated surface, but had little-to-no effect when stepping on the uncoated surface (*Figure 5c*; *Video 4*; *Figure 5—figure supplement 1j–o*; *Supplementary file 3*). No differences in the base-of-support were detected between the two surfaces, suggesting that balance/postural changes likely do not account for this intriguing result (18.36 ± 2.97° vs 21.44 ± 4.48°; p>0.05, paired t-test).

We then explored the effects of LAPN silencing on left-right hindlimb coordination in a different environmental context: water. Swimming is a bipedal task where the hindlimbs provide the major propulsive force while the forelimbs occasionally steer (*Gruner and Altman, 1980*). As the limbs are unloaded, both proprioceptive and cutaneous feedback associated with plantar stepping is altered (*Akay et al., 2014*). In

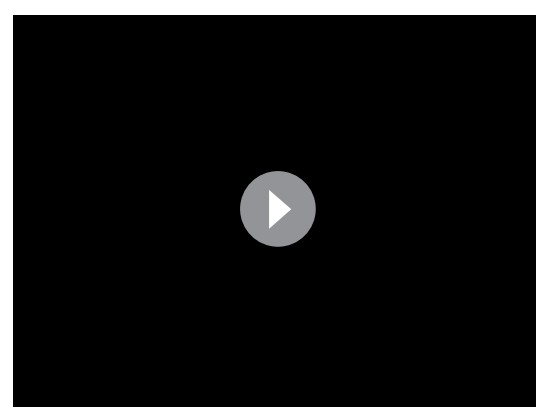

**Video 2.** Silencing long ascending propriospinal neurons (LAPNs) disrupts interlimb coordination during overground stepping but not during treadmill-based locomotion. Videos shown from the same animal at 1x, 0.5x, and 0.25x speed during overground and treadmill stepping at Dox1$^{On}$Days 4 and 5.
https://elifesciences.org/articles/53565#video2

contrast to our overground findings, silencing LAPNs had no effect on left-right hindlimb alternation during swimming (*Figure 5d*; *Video 5*; *Figure 5—figure supplement 1p*; *Supplementary file 3*), further supporting the concept that LAPNs help secure interlimb coordination in a context-dependent manner.

Finally, we explored if the context-specificity of silencing-induced disruptions to interlimb coordination was related to speed and speed-related gait changes. We discovered that silencing modestly expanded the speed ranges expressed when speed was determined by the animal (overground, exploratory, coated and smooth). However, the vast majority of disrupted steps occurred at speeds that were shared across the behavioral contexts examined, whether there were few (treadmill, exploratory and smooth surface) or many (overground and coated; *Figure 5e*, *Figure 5—figure supplement 2*). Using the data generated from age-matched control rats on the three meter long runway (*Figure 4a*, *Figure 4—figure supplement 1*) we found, as expected, a strong relationship between speed and hindlimb coordination (data not shown; Spearman Rank correlation coefficient = 0.753, N = 12 age-matched control rats, n = 403 total steps analyzed). In contrast, when we ran a similar comparison for the Dox$^{On}$ overground stepping data, we could find no predictable relationship (Spearman Rank correlation coefficient = 0.410, N = 13, n = 600 total steps analyzed) suggesting that silencing the LAPNs resulted in interlimb coordination disruptions that were not speed-dependent. Overall, these data show that disrupted steps occurred throughout the speed range regardless of behavioral context, and that the majority occurred at speeds ($\leq$90 cm/s) normally associated with walk-trot (alternating gaits; *Figure 5—figure supplement 2ee*), illustrating that silencing-induced changes to interlimb coordination were not related to speed-dependent gait change.

## Discussion

Given their lumbar-to-cervical connectivity, we hypothesized that silencing LAPNs would disrupt hindlimb-forelimb coordination during locomotion. Instead, we unexpectedly uncovered a role for the LAPNs in securing left-right limb alternation at each girdle. The other salient features of locomotion remained wholly intact, including intralimb coordination, balance/posture, the overall 1:1 step ratio, the fundamental relationship between speed and the spatiotemporal features of limb movement, and the underlying locomotor rhythm. Collectively, these findings suggest LAPNs reside within the interlimb pattern formation layer of the locomotor hierarchy and are functionally separate from the circuitry responsible for the underlying rhythm and intralimb coordination. Interestingly, these outcomes dovetail with previous work where spinal L2 interneurons that project to L5 were silenced (*Pocratsky et al., 2017*). In that case, hindlimb alternation was selectively disrupted, allowing a spectrum of coupling patterns to be expressed, while other essential features of locomotion were once again preserved (*Pocratsky et al., 2017*). Together, these studies indicate that inter-segmental projecting lumbar pathways are key distributors of temporal information that can be used for maintaining left-right alternation during overground locomotion, and that hindlimb-forelimb coordination is either secured by other means or is less vulnerable to disruption potentially requiring silencing of larger numbers or a wider range of long propriospinal neurons.

It is generally accepted that left-right coordinating circuits are functionally organized into gait-specific ensembles, each recruited as a function of speed (*Deska-Gauthier and Zhang, 2019*). In the walking ensemble where the limbs move at low speed, left-right alternation is governed through a distributed network of ventrally-derived, commissural-projecting inhibitory spinal neurons (the "V0d" class) (*Bellardita and Kiehn, 2015*). As speed increases, the trotting ensemble is recruited wherein faster-paced left-right alternation is primarily secured through the combined actions of the excitatory V0 neuronal subclass ("V0v" spinal neurons) (*Bellardita and Kiehn, 2015*; *Talpalar et al., 2013*) and the excitatory, ipsilateral-projecting "V2a" subclass (*Crone et al., 2009*; *Crone et al., 2008*). At this time the circuits comprising the bounding ensemble remain largely unknown (*Deska-Gauthier and Zhang, 2019*). Through this modular organization, distributed classes of spinal interneurons are recruited as a function of speed, ensuring that appropriate patterns of limb coordination are expressed for each gait. By leveraging the spatial (anatomically defined) and temporal (inducible on-off) aspects of conditional silencing, we have highlighted an underlying complementary feature to the modular control of locomotion: flexibility. When the LAPNs were silenced, the system was able to accommodate right-left coupling patterns normally associated with the high-speed gaits of gallop and bound, but at walking speeds. This intrinsic freedom of pattern expression across a range of

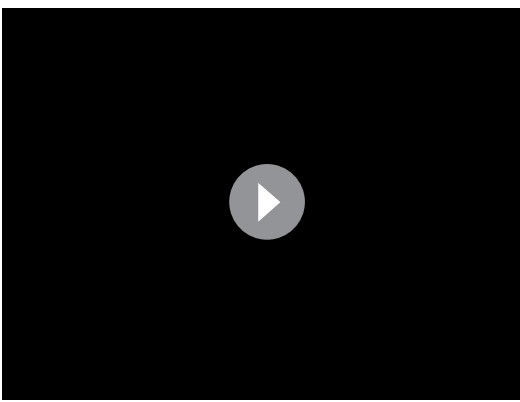

**Video 3.** Interlimb coordination is not affected during exploratory-like stepping behavior. Videos shown from the same animal at the same Dox^On time point at 1x, 0.5x, and 0.25x speed.

https://elifesciences.org/articles/53565#video3

stepping speeds exposes a more flexible organization schema for locomotor control, a key tenet for adaptability in motor behavior.

Beyond the unexpected observation that silencing an ascending inter-enlargement pathway partially decouples the left-right limb pairs at each girdle, our most intriguing result is that the silencing phenotype is context-dependent. What could account for this striking phenomenon? A parsimonious interpretation would be that LAPNs are necessary for securing interlimb coordination in select conditions, such as overground stepping ("going from A to B") on a surface with good grip like Sylgard. Conversely, in other conditions such as stepping overground on uncoated acrylic (or during exploratory-like stepping, stepping on a treadmill, or swimming), LAPNs are dispensable. This rigid supposition may appear untenable given the rich repertoire of behaviors expressed by mammalian spinal circuitry. Thus, we offer an alternative interpretation by speculating that there exists a dynamic relationship between spinal autonomy and supraspinal oversight. Classic studies in the cat show that the lumbar spinal cord can produce the fundamental rhythm and pattern of locomotion even in the absence of all supraspinal and sensory input (*Grillner, 1981*). So, in a more "spinal autonomous" context (e.g. non-exploratory nose-up stepping overground "from A to B" on a Sylgard-coated surface), LAPNs are critical for limb-pair coupling such that their conditional silencing disrupts intra-girdle alternation and this disruption is not "corrected" by supraspinal (or any other) oversight. When the terrain changes (uncoated Plexiglas) or when stepping on a treadmill (*Shefchyk et al., 1984*), functionally parallel pathways may be engaged that would ensure a stable pattern of intra- and inter-girdle movements, thereby masking or temporarily over-riding the functional consequence(s) of silenced LAPNs. During exploratory (*Sinnamon, 1993*) behavior (nose-down) we observed very precise alternation of both the forelimbs and hindlimbs, and very precise hindlimb-forelimb coordination (*Figure 5—figure supplement 1*) that was not disrupted even slightly during LAPN silencing, arguing perhaps that exploratory stepping appropriate for olfaction or whisking involves a very stable pattern relying strongly on afferent input as dictated by the

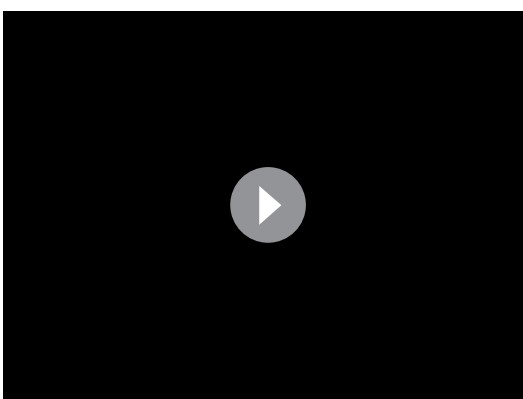

**Video 4.** Silencing long ascending propriospinal neurons (LAPNs) selectively disrupts interlimb coordination when animals are locomoting on a coated, but not smooth stepping surface. Videos shown from the same animal at the Control and Dox^On time points at 1x and 0.5x speed.

https://elifesciences.org/articles/53565#video4

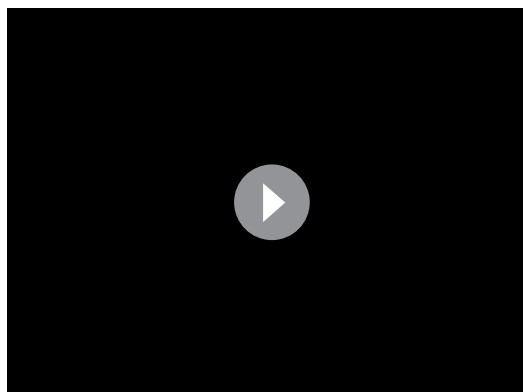

**Video 5.** Silencing long ascending propriospinal neurons (LAPNs) does not disrupt left-right hindlimb alternation during swimming. Videos shown from the same animal at 1x, 0.5x, and 0.25x speed.

https://elifesciences.org/articles/53565#video5

needs of supraspinal centers. Thus, the functional importance of LAPNs for securing interlimb coordination would rise or fall depending on the behavioral context or environmental conditions, which we interpret as decreased or increased supraspinal oversight. Nonetheless, this hypothesis is still parsimonious in that it does not take into account state-dependent neuromodulation of motor networks, a powerful phenomenon wherein circuits are reconfigured to produce needed frequencies and phase relationships (*Marder et al., 2015*). Swimming, which is primarily hindlimb-driven, may actually be a "lumbar autonomous" activity thus rendering LAPNs, and the information they carry, dispensable.

Based on the context-specificity of the phenotype, it is logical to conclude that the LAPNs carry temporal information from the hindlimb locomotor circuitry to the forelimb locomotor circuitry. However, the source of that temporally modulated information is unclear. It might be derived primarily from intrinsic spinal circuitry that generates the underlying rhythm of stepping (the rhythm-generating layer) and that is separate from, but has influence over, limb alternation at each girdle. Alternatively, it might be derived principally or entirely from hindlimb afferent input carrying temporal information associated with paw contact (cutaneous), limb loading or joint movement (proprioceptive). However, our previous work utilizing conditional synaptic silencing suggests that this alternative may be incorrect. When we silenced L2 interneurons that project to L5, we selectively disrupted hindlimb alternation, which should, in turn, have altered any temporal information derived from the hindlimb movement being carried rostrally by LAPNs. However, apart from alternation, no disruptions to forelimb function or any other salient features of stepping were observed (*Pocratsky et al., 2017*). Thus, the temporal information carried by the LAPNs may arise from the rhythm-generating circuitry, as suggested earlier, or may be some derivative of output from the intrinsic spinal circuitry (rhythm and pattern) and sensory input. Ultimately, the mechanisms underlying these striking results remain unknown. Computational modeling of the intrinsic and extrinsic network dynamics will be required to shed light on this phenomenon.

In conclusion, by reversibly silencing LAPNs in the otherwise intact adult rat, we show that a stable locomotor rhythm and intralimb pattern is maintained even while alternation, a key feature of the walk and trot gaits, is disrupted. We observed a wide range of coupling patterns expressed concomitantly with overall network stability. These observations highlight a surprising flexibility within locomotion and the spinal circuitry that governs it.

## Materials and methods

### Key resources table

| Reagent type (species) or resource | Designation | Source or reference | Identifiers | Additional information |
|---|---|---|---|---|
| Strain, strain background (female Sprague-Dawley rats) | | Envigo | | 200–220 g, approximately 10–12 weeks old |
| Antibody (GFP) | Rabbit IgG | Abcam ab290 | | 1:5000 |
| Antibody (NeuN) | Guinea pig IgG | Millipore ABN90P | | 1:500 |
| Antibody (NeuN) | Mouse IgM | Millipore MAB377 | | 1:500 |
| Antibody (neurofilament) | Mouse IgM | Sigma N5264 | | 1:30,000 |
| Antibody (synaptophysin) | Mouse IgM | Millipore MAB5258-50UG | | 1:10,000 |
| Antibody (vesicular glutamate transporter 2) | Guinea pig IgG | Millipore AB2251-I | | 1:5000 |
| Antibody (vesicular GABA transporter) | Goat igG | Frontier Institute VGAT-Go-Af620 | | 1:500 |

*Continued on next page*

*Continued*

| Reagent type (species) or resource | Designation | Source or reference | Identifiers | Additional information |
|---|---|---|---|---|
| Antibody (non-immune sera) | Rabbit IgG | Jackson ImmunoResearch #711-005-152 | | 1:5000 |
| Antibody (secondary AlexaFluor 488) | Rabbit IgG | Jackson ImmunoResearch # 711-545-152 | | 1:200 |
| Antibody (secondary AlexaFluor 594) | Guinea pig IgG | Jackson ImmunoResearch #706-585-148 | | 1:200 |
| Antibody (secondary AlexaFluor 594) | Mouse IgG | Jackson ImmunoResearch # 715-585-150 | | 1:200 |
| Antibody (secondary AlexaFluor 647) | Mouse IgG | Jackson ImmunoResearch # 715-605-151 | | 1:200 |
| Antibody (secondary AlexaFluor 647) | Guinea pig IgG | Jackson ImmunoResearch # 706-546-148 | | 1:200 |
| Antibody (secondary AlexaFluor 647) | Goat IgG | Jackson ImmunoResearch # 705-605-147 | | 1:200 |
| HiRet-TRE-EGFP.eTeNT | | Generous gift from Tadashi Isa | | $1.6 \times 10^7$ vp/ml |
| AAV2-CMV-rtTAV16 | | Generous gift from Tadashi Isa | | $4.8 \times 10^{12}$ vp/ml |
| HiRet-Cre | | Generous gift from Zhigang He | | $1.6 \times 10^{12}$ vp/ml |
| AAV2-CAG-FLEx-GFP | | UNC Vector Core | | $3.5 \times 10^{12}$ vp/ml |
| Chemical compound (Sylgard) | Sylgard-coated surface | Sylgard 184 Silicone Elastomer Kit, Dow Corning | | |
| Chemical compound (cholera toxin B subunit conjugate) | CTB-488 | Invitrogen/Molecular Probes C-34775 | | 1.5% solution in sterile saline |
| Chemical compound (cholera toxin B subunit conjugate) | CTB-594 | Invitrogen/Molecular Probes C-34777 | | 1.5% solution in sterile saline |
| Chemical compound (cholera toxin B subunit conjugate) | CTB-647 | Invitrogen/Molecular Probes C-34778 | | 1.5% solution in sterile saline |

Experiments were performed in accordance with the Public Health Service Policy on Humane Care and Use of Laboratory Animals, and with the approval of the Institutional Animal Care and Use and Institutional Biosafety Committees at the University of Louisville.

A total of N = 45 adult female Sprague-Dawley rats (Envigo; 200–220 g, approximately 10–12 weeks of age) were used throughout this study. Animals were housed two per cage under 12 hr light:dark cycle with ad libitum food and water. Power analysis of previous silencing experiments revealed that N = 6 was sufficient to detect a significant difference in behavioral outcome measures with 90–99% power (*Pocratsky et al., 2017*). Silencing data shown in *Figures 2–5* represent two

separate experiments, each N = 6 and N = 7, respectively. Experiments were performed in a staggered fashion separated by one month such that when the first group was undergoing Dox2 testing, the second group was performing Dox1 testing. No significant differences were detected between the two groups. Data shown are from the pooled samples (N = 13).

### Viral vector production

Dr. Tadashi Isa and colleagues generously provided the plasmid vectors (*Kinoshita et al., 2012*). The HiRet-TRE-EGFP.eTeNT and AAV2-CMV-rtTAV16 viral vectors were built following previously described methods with viral titers of $1.6 \times 10^7$ vp/ml and $4.8 \times 10^{12}$ vp/ml, respectively (*Pocratsky et al., 2017*; *Abdellatif et al., 2006*; *Sommer et al., 2003*).

### Intraspinal injections to double infect and silence LAPNs

Intraspinal injections were performed as described previously (*Pocratsky et al., 2017*). Procedural details have been deposited into the Nature Protocol Exchange (http://dx.doi.org/10.1038/protex.2017.125).

We adapted this protocol to target LAPNs by performing a C6-C7 laminectomy to expose spinal C6 and injected HiRet-TRE-EGFP.eTeNT using coordinates of 0.6 mm mediolateral and 1.3 mm dorsoventral. The AAV2-CMV-rtTAV16 viral vector was similarly injected into L2 at 0.6 mm mediolateral and 1.5 mm dorsoventral. In double-infected neurons that constitutively express rtTAV16, doxycycline (DOX) induces enhanced tetanus neurotoxin (eTeNT) expression. eTeNT is then transported to the terminal field where it prevents exocytosis of synaptic vesicles, thereby silencing neurotransmission. Removing DOX from the drinking water restores neurotransmission, allowing acute and reversible silencing of this anatomically defined pathway in the otherwise intact adult rat.

### LAPN silencing experimental timeline

The experimental design used is similar to that of our previous silencing experiments (*Pocratsky et al., 2017*). In addition to the previously described control and Dox time points, we included an additional vehicle control (sucrose water without doxycycline). N = 6 animals underwent behavioral testing following 4 days of sucrose water. No significant differences were detected between the Sugar control and all other control (or Dox) time points.

Animals were acclimated to the stepping chamber prior to Baseline acquisition. All stepping behavior analyzed was spontaneous and volitional. Animals did not receive positive or negative reinforcement training. Only the walk-trot gait was observed at control time points (no spontaneous galloping or bounding was seen). The order in which animals were tested was random. Raters were blinded to animal-specific behavior across time points and behavioral tasks. Each animal served as its own control throughout the study as previously described (*Pocratsky et al., 2017*).

Unless otherwise stated, control data reflect the combined data from the following time points: Baseline, Pre-Dox1, Dox$^{Off}$, and Pre-Dox2. Similarly, the Dox data reflect the combined data from the following time points: Dox1$^{ON}$Day 3 ("-D3"), -D5, -D8 and Dox2$^{ON}$D3, and -D5. Unless otherwise stated, "Control" refers to collapsed data from all control time points (excluding sugar control) and "Dox" refers to collapsed data from all Dox time points.

### Hindlimb kinematics and intralimb coordination analyses

Hindlimb kinematic analyses were performed as previously described (*Pocratsky et al., 2017*; *Kuerzi et al., 2010*; *Magnuson et al., 2009*) , using custom-built Excel add-in macros (*Morehouse, 2020*; copy archived at https://github.com/elifesciences-publications/KSCIRC-Gait-Addin).

### Overground locomotion analyses

The overground locomotor analysis was performed following previously described methods and inclusion/exclusion criteria (*Pocratsky et al., 2017*). Data were analyzed with and without speed as a co-variate.

To calculate the magnitude change in interlimb coordination during LAPN silencing, we first calculated the number of altered steps (beyond control variability) for each animal for Control and Dox time points for the following limb pairs: left-right forelimb, left-right hindlimb, right homolateral limb pair ("ipsi hindlimb-forelimb"), and right hindlimb-left forelimb pair ("contra hindlimb-forelimb").

After calculating the total number of altered steps for each animal (in the analyzed locomotor bouts), we determined the percent of disrupted steps for left-right or hindlimb-forelimb limb pairs.

To calculate the group peak effect of LAPN silencing, we first identified the Dox time point that showed peak changes to interlimb coordination. We stratified the animals into either Dox1 or Dox2 categories and then performed comparisons (see Statistics section below). One animal did not show changes in left-right hindlimb coordination (*Figure 2—figure supplement 1e*, filled circles), but did show silencing-induced perturbations to left-right forelimb and contralateral hindlimb-forelimb coordination.

Interlimb coordination (phase)-frequency polar plots were created in SigmaPlot (ver 22) with each concentric circle set to 2 Hz increments (inner most: 0 Hz, outer most: 10 Hz). All steps analyzed (Control, n = 480; Dox, n = 600) were plotted for the raw left-right coordination value and its associated step frequency value. The dashed circle denotes a 5 Hz threshold at which almost all Control steps fell within (forelimbs: 99.8% of all steps; hindlimbs: 100%). Data were compared for the circular dispersion as described below (Statistics section). Phase-frequency polar plots were similarly created for the speed-dependent gaits (see the "Volitionally-expressed, speed-dependent gaits" section below for experimental details).

The underlying rhythm indices were analyzed as described previously (*Pocratsky et al., 2017*). Briefly, we first confirmed that there were no significant differences between the left and right limbs at Control and Dox time points, respectively. We then calculated the average stride duration for the fore- and hindlimbs, respectively. We also compared between the limb pairs for Control and DOX as well (bars: group mean ± S.D.; circles: individual means). Regression and slope analyses were performed (comparing Control vs Dox) on the following: left versus right forelimb stride duration, left versus right forelimb stride frequency, left versus right hindlimb stride duration, left versus right hindlimb stride frequency, forelimb versus hindlimb stride duration, and forelimb versus hindlimb stride frequency. The inter-girdle comparisons had the left and right limb pairs averaged together before hindlimb versus forelimb analyses.

## Postural stability

Balance, posture, and trunk control were assessed through a series of graded tasks. Base-of-support analyses were focused on the hindlimbs as this is the site wherein the major propulsive forces for locomotor behaviors are generated. Using a three-point angle model (point 1: area between shoulder blades, 2: groin, 3: hind paw position at initial contact), the rotation of the hind paws at initial contact were quantified for each step cycle. We chose to use the initial contact instead of lift-off as there is some normal rotation of the paw as weight is differentially transferred to the hindlimb throughout the stance phase. Both the left and right hindlimbs were analyzed at Baseline (n = 220–227 total step cycles analyzed per left or right hindlimb for N = 13 animals) and Dox1$^{On}$-D5 (n = 223–229 total step cycles).

Animals were tested on the horizontal ladder (Columbus Instruments; Columbus, OH, USA, 2.5 mm rungs spaced 3.5 cm apart) (*Burke et al., 2012*) during the following time points: Baseline, Pre-Dox1, Dox1$^{On}$-D4, Dox1$^{On}$-D8, Dox$^{Off}$, Pre-Dox2, Dox2$^{On}$-D4, and Dox2$^{On}$-D5. Each animal underwent five stepping trials per time point. The total number of footfalls were quantified for the left and right hindlimbs, respectively, for each animal at each time point. As no statistical difference between the left and right hindlimbs was observed, we combined the trials for the left and right limbs and determined each animal's overall average number of footslips for Control and Dox, respectively. Statistics were performed on the group means (bars: average ± S.D.; circles: individual means overlaid). There was one outlier in the data set (red circle;>4 s.D.). Excluding the outlier from analyses did not change the results (Control mean: 3.33 ± 2.4 with outlier, 2.70 ± 1.02 without outlier; both p<0.001 when compared to Dox [1.09 ± 0.54]).

Animals traversed a custom-built 1.8 cm wide beam during the following time points: Baseline, Pre-Dox1, Dox1$^{On}$-D3, Dox1$^{On}$-D5, Dox1$^{ON}$-D8, Dox$^{Off}$, Pre-Dox2, Dox2$^{On}$-D3, and Dox2$^{On}$-D5. Each animal underwent three beam walk trials per time point assessment. The total number of foot falls from each trial per animal per time point for the left and right hindlimbs, respectively, were calculated. As no significant difference between the left and right sides was detected, we combined the trials for both hindlimbs and calculated the average number of footfalls for Control and Dox, respectively, for each animal. Statistical analyses were performed on the group means. Excluding the

outlier shown in *Figure 3j* (red circle) yielded similar results. Animals also stepped on beams with a width of 3.6 cm and 5.4 cm, respectively, and showed little-to-no footfalls (data not shown).

Sagittal recordings of animals in the stepping chamber were analyzed for volitional rearing. We defined rearing as when the animal fully supported itself on its hindlimbs only (grooming events excluded). We defined the onset of rearing as when the animal removed its last forepaw from the ground (removal of all digits). The completion of the rearing event was defined as when a forepaw returned to the ground. We quantified the frequency and duration of all spontaneously expressed rearing events for all animals across all time points. To stratify the rearing events based on the level of forepaw support, we documented the onset times of when the forepaw contacted the side of the acrylic chamber, came into visual focus, and demonstrated weight bearing through spreading of fingertips and postural adjustments. The completion of forepaw support was defined as when the paw was removed from the glass as seen by postural movements, blurring of the hand, and narrowing of the fingertips. As such, we could define the degree of forepaw support by both frequency and duration of the events. Any event where the forepaws were out the field of view were excluded from analysis. The overall average frequency and duration of spontaneously evoked rearing bouts were calculated for each animal across all Control and Dox$^{On}$ time points, respectively.

The trunk angle (degree at which the animals held their bodies relative to the water surface) was calculated using a four-point angle model (points 1 and 2: water surface [left and right extremes of the videos], 3: iliac crest; 4: hip). The trunk angle was calculated throughout the stroke cycle on a stroke-by-stroke basis for each swimming pass. Data shown are from Pre-Dox1 and Dox1$^{ON}$-D5 with a total of n = 7873 and n = 10,520 trunk angles analyzed, respectively, for each hindlimb per animal. Data shown are the group mean ± S.D. (circles denote individual animal means).

## Generalized behavioral analyses: context is key

Treadmill-based locomotion (Single Lane Gait Analysis Treadmill, Columbus Instruments; Columbus, OH, USA) was analyzed following previously described methods (*Pocratsky et al., 2017*; *Beare et al., 2009*). Treadmill testing was performed at the following time points: Baseline, Pre-Dox1, Dox1$^{On}$-D4, Dox$^{Of}$, Pre-Dox2, and Dox2$^{ON}$-D4. Inclusion criteria for the steps analyzed including the following: locomotor bouts where animals (1) consistently stepped in the middle of the treadmill, (2) did not hesitate/pause and "ride" to the back of the enclosure, (3) had minimal lateral deviations during stepping, and (4) did not have forward propulsive actions from the end of the enclosure to the middle and/or front. Recordings were analyzed using the MaxTRAQ software package (Innovision Systems Inc; Columbiaville, MI, USA). Care was taken to minimize the number of stepping sessions due to the adverse training effects associated with increased exposure to treadmill stepping (*Beare et al., 2009*; *Hamers et al., 2006*). We observed no instances where the animals spontaneously bounded (half or full) on the treadmill (N > 430 steps).

We noticed that when animals were 'exploring' their environment (e.g. snout in close proximity to the ground during locomotor bout), the silencing phenotype was absent. However, if the animals were stepping across the walkway chamber with no distractions, the phenotype was expressed. For descriptive purposes, we have termed these two behaviors as exploratory and non-exploratory stepping "modes." To analyze the effects of LAPN silencing during these two behavioral conditions, we applied strict criteria to the analyses of exploratory stepping. Using sagittal recordings as the reference, the following inclusion criteria were applied: (1) animals must have their snouts pointed downwards throughout the entirety of the step sequence, (2) animals must step consistently with no pauses or hesitations at any moment throughout the locomotor bout, (3) animals must step across at least ¾ the walkway, and (4) animals must locomote with little-to-no lateral deviations. Every animal displayed some form of "snout down" exploratory behavior at a Control and Dox$^{On}$ time point, respectively. A total of n = 100 and n = 95 step cycles were analyzed across all Control and Dox time points, respectively. The non-exploratory stepping data are shown from that in *Figures 2–4*.

The influence of the stepping surface was discovered in a separate LAPN silencing study. N = 8 adult female Sprague-Dawley rats (215–225 grams) received the aforementioned viral vector injections with behavioral testing performed at Baseline, Pre-Dox1 (approximately 3 weeks post-injections), Dox1$^{On}$-D5, Dox1$^{On}$-D8, and Dox$^{Off}$. In this study, animals were tested in two acrylic walkway chambers with different stepping surfaces. One walkway was coated with a clear, silicone substance ("coated"; coefficient of friction = 1.41) (Sylgard 184 Silicone Elastomer Kit; Dow Corning; Midland, MI, USA) while the other walkway was uncoated acrylic (coefficient of friction = 0.47). A total of 10–

12 step cycles were analyzed for each animal across all time points. The control threshold (average + 2 s.D.) was calculated for each stepping surface, respectively, from data generated at Baseline, Pre-Dox1, and Dox$^{Off}$. No significant differences were detected between the stepping surfaces at control time points.

The coefficients of friction reported for each stepping surface were calculated using the following approach. First, an alert adult female Sprague-Dawley rat (229 grams) was positioned into one side of the stepping chamber. While the animal calmly rested, the tank was slowly raised until paw traction was lost. This angle was measured in three separate trials for both the Syglard-coated and uncoated acrylic tanks, respectively. The coefficient of friction was then calculated based on the average of the tangent of the three measured angles. This process was repeated with an object that closely approximates the texture of the paw surface (e.g. smooth wooden block), yielding similar coefficients for each surface (uncoated acrylic: 0.44, Sylgard coated: 1.73).

Hindlimb swim analyses were performed following previously described methods (*Pocratsky et al., 2017*).

## Volitionally-expressed, speed-dependent gaits

To interpret our silencing data with respect to the speed-dependent locomotor gaits, we had to devise a strategy that would allow animals to freely express these fast-paced gaits overground, but still be compatible with our in-house methods for gait analyses (ventral recordings). To do this, we designed and built a runway chamber ("long tank") that was 305 cm long, 30.5 cm wide, and 14 cm tall with four high-speed video cameras (200 Hz) evenly spaced beneath the stepping surface.

To "stitch" together the multiple cameras such that all steps could be accounted for across the length of the tank, we used the following strategy. First, we arranged the cameras such that the FOV overlapped (e.g. camera 1–2, 2–3, 3–4). We placed two markers (between the first and second as well as the third and fourth cameras) to use as points of reference during video analysis. These points were copied to all videos such that the stepping coordinates were integrated across the four individual files acquired (one per camera). Using these strategies, we had no missing frames or steps when animals stepped between the different FOVs. To prevent or "subtract out" digitization of steps that fell within two FOVs, we created a series of inter-camera markers throughout the length of the tank. We measured the distance between the start of the tank to each of these markers and quantified these points during video analysis. Thereafter, we custom built a macro that would detect these digitized inter-camera markers to then filter out the "extra digitizing" between two overlapping FOVs. These processes were also repeated for cameras three and four. Each camera has a 5 cm scale visible, allowing us calibrate the video files using the MaxTRAQ scale feature. Within our macro, we created a pixel-to-cm conversion factor that allowed us to reliably measure the various spatiotemporal indices of locomotion. This experimental design allowed us to stitch together multiple videos for seamless step analyses.

Given the length of the tank, we found that animals were often distracted and rarely completed one complete locomotor bout without pausing to explore. To combat this, we devised a training program that included positive reinforcement to encourage the completion of a locomotor bout across the 3 m tank. N = 12 naïve adult female Sprague-Dawley rats (200–220 g) underwent this training program to generate the speed-dependent gait data. Details of this program are as follows.

First, animals were extensively handled by the experimenters to where they would freely approach and climb into the researchers outstretched hand. During these gentling sessions, animals were handled individually and/or with their cage mates and provided positive reinforcement after they were returned to their home cage (food reward). Once animals were well-acclimatized to the experimenters, they were then introduced to the long tank. Cage mates were placed in chamber together, allowing them to freely explore and run/play throughout the full length of the tank. Food rewards were provided at each end of the tank during these sessions such that animals began to associate these areas with treats. This phase of the training program lasted approximately 3–4 days.

After the initial introduction phase to the long tank, animals began extensive training where they were encouraged to step across the entire length of the runway with little-to-no pausing/hesitations. During these training sessions, two experimenters were positioned at either end of the tank. To start, one trainer would create a sound (gently tapping the side of the tank or lightly rubbing two gloved fingers back and forth). Animals typically stepped towards the side of the stimuli where they received a food reward. Thereafter, the second experimenter would provide auditory stimuli and the

animal would turn around to fully traverse the tank again to receive another food reward. No food reward was given if the animal did not successfully complete one pass start to finish (no pausing, no hesitations). (This phase of the training program lasted two weeks with twice a day training sessions during the first week and one training session per day during the second week). By the end of these training sessions, animals freely expressed their natural repertoire of gaits (walk-trot, gallop, half-bound, and full-bound), sometimes even to the sound of the experimenter rubbing their gloves.

Data shown are from seven separate recording sessions that were spread out over a fourth month period. Food rewards were not given during the video recording sessions. However, the experimenters did provide the auditory stimuli to which the animals were accustomed during training. Our defining criteria for the distinct locomotor gaits are based on previously described coupling patterns (*Bellardita and Kiehn, 2015*). We did not distinguish between the two alternating gaits: walk (three limbs in contact with the ground) and trot (two limbs in contact with the ground at any moment). A total of n = 160, 50, 108, and 80 step cycles were analyzed for the walk-trot, gallop, half-bound, and full-bound gaits, respectively. Fewer gallop step cycles were analyzed due to the transient nature of this gait (*Lemieux et al., 2016*).

We did not test for the expression of the speed-dependent gaits in the long tank during LAPN silencing. In the long tank paradigm, we applied positive reinforcement to encourage the volitional expression of the faster-paced gaits. These gaits are volitional in the sense that the animals were not placed on a treadmill and "forced" to step a fast rates of speed. During silencing, we did not want to confound our results by "encouraging" the expression of distinct coupling patterns. It would be challenging to reconcile whether changes in the coupling patterns expressed were due to LAPN silencing or the reinforcement of fast-paced gait expression. Instead, we first wanted to assess how the nervous system would intrinsically respond to the "functional loss" of LAPNs. Going forward, these data may serve as a foundation for future experiments where the gaits are systematically assessed during conditional silencing.

## Histological processing for double-infected LAPNs

Following terminal assessments, animals were sacrificed with an overdose of ketamine:xylazine and transcardially perfused with 0.1 M PBS (pH 7.4) followed by 4% paraformaldehyde (PFA). Spinal cords were dissected, post-fixed in 4% PFA for 1 to 3 hr, and transferred to 30% sucrose for 3–4 days at 4°C. The cervical and lumbar injection sites were dissected, embedded in tissue- freezing medium, cryosectioned at 30 μm in five sets, and stored at −20°C.

Immunohistochemical detection of EGFP.eTeNT-positive terminals in the caudal cervical segments was performed following previously described methods (*Pocratsky et al., 2017*) (Nature Protocol Exchange: http://dx.doi.org/10.1038/protex.2017.141). Antibodies used include the following: rabbit anti-GFP (abcam ab290, 1:5,000), guinea pig anti-NeuN (Millipore ABN90P, 1:500), mouse anti-NeuN (Millipore MAB377, 1:500), mouse anti-neurofilament (Sigma N5264, 1:30,000), mouse anti-synaptophysin (Millipore MAB5258-50UG, 1:10,000), guinea pig anti-vesicular glutamate transporter 2 (Millipore AB2251-I, 1:5,000), and goat anti-vesicular GABA transporter (Frontier Institute VGAT-Go-Af620, 1:500; see manufacturer for validation details). Negative controls include non-immune sera matched for protein concentration and dilution (donkey anti-rabbit IgG; Jackson ImmunoResearch #711-005-152, 1:5,000). Secondary antibodies were used at a dilution of 1:200 and included the following (all donkey host): anti-rabbit IgG AlexaFluor 488 (Jackson ImmunoResearch # 711-545-152), anti-guinea pig IgG AlexaFluor 594 (Jackson ImmunoResearch #706-585-148), anti-mouse IgG AlexaFluor 594 (Jackson ImmunoResearch # 715-585-150), anti-mouse IgG AlexaFluor 647 (Jackson ImmunoResearch # 715-605-151), anti-guinea pig IgG AlexaFluor 647 (Jackson ImmunoResearch # 706-546-148), and anti-goat IgG AlexaFluor647 (Jackson ImmunoResearch # 705-605-147). The applied microscopy settings and post hoc image processing are previously described (*Pocratsky et al., 2017*).

Double-infected LAPNs were detected following methods previously described (Nature Protocol Exchange: http://dx.doi.org/10.1038/protex.2017.142). In light of the reduced post-fixation time (1–3 hr vs overnight), the following modifications were applied: (1) antigen retrieval was excluded, (2) rabbit anti-GFP was used at a range of 1:30,000 to 1:60,000 to amplify endogenous eTeNT.EGFP signal, and (3) the blocking, secondary, and streptavidin HRP steps were each 30 min in duration. For a negative control for GFP, an isotype-matched IgG at identical protein concentration and

dilution was used (donkey anti-rabbit IgG; Jackson ImmunoResearch #711-005-152). The microscopy settings and post hoc image processing are previously described (*Pocratsky et al., 2017*).

## LAPN anatomical work-up
### CTB Labeling
Power analyses revealed that a sample size of N = 5 animals was sufficient to detect a significant difference in the number of ipsilateral-projecting vs contralateral-projecting rostral LAPNs, with or without local projections to spinal L1 or spinal L5, respectively (power >95%). A total of N = 11 adult female Sprague-Dawley rats (210–230 grams) were used in this study, with N = 5 and N = 6 comprising two separate groups (described below).

### Cervical and lumbar injections were performed during the same day of surgery
Animals were anesthetized with a cocktail of ketamine/xylazine/acepromazine (80 mg/kg, 4 mg/kg, and 305 mg/kg; i.p.) and received a C6-C7 laminectomy to expose spinal C6. Following previously described methods, two different AlexaFluor conjugates of cholera toxin B subunit (CTB) were bilaterally injected into the intermediate gray matter (*Pocratsky et al., 2017*). Animals received one unilateral injection of CTB-AlexaFluor-594 on the left field of view (FOV) and one unilateral injection of CTB-AlexaFluor-647 on the right field of view. Injection coordinates were 0.5 mm mediolateral and 1.3 mm dorsoventral, respectively. Following the cervical injections, animals were randomly assigned to two groups. One group received a T12 laminectomy to expose spinal L1 (N = 6) while the second group received a ~ T13 L1 laminectomy to expose spinal L5 (N = 5). Both groups received one unilateral injection of CTB-AlexaFluor-488 (right FOV). The L1 injections were performed at the rostral FOV with mediolateral-dorsoventral coordinates of 0.5 mm and 1.3 mm, respectively. The L5 injections were performed at the caudal FOV with mediolateral-dorsoventral coordinates of 0.5 mm and 1.4 mm, respectively. All CTB conjugates were prepared as a 1.5% solution (0.1 M PBS, pH 7.4; Molecular Probes, Eugene, OR, USA) and delivered in two, 0.25 µl boluses separated by three minutes to allow for tracer uptake. Post-operative care was performed as described above.

Three weeks later, animals were euthanized and the spinal cords were dissected, post-fixed for one hour, and then stored at 4°C in a 30% sucrose solution. To analyze retrogradely-labeled LAPNs, spinal T13-L6 was dissected, embedded in tissue-freezing medium, and cryosectioned at 20 µm in sets of five (adjacent sections separated by 100 µm rostrocaudally). To analyze the cervical injection sites, spinal C5-C8 was dissected, embedded, and serially cryosectioned at 30 µm. All sections were mounted onto charged glass slides and stored at −20°C.

Power analyses revealed that N = 5–7 sections/animal were needed to detect a significant difference in the number of ipsilateral-projecting versus contralateral-projecting rostral LAPNs, positive or negative for L1 or L5 local collaterals (power >82%). Proportional cell counts of LAPNs with L1 or L5-projecting collaterals were performed as previously described (*Pocratsky et al., 2017*). A total of n = 6,775 LAPNs were counted across N = 11 animals. Careful attention was paid to the in vivo injection site FOVs for the schematics shown as well as projection pattern identification (ipsi- vs contralateral). Representative images are shown. Data shown are proportional cell counts of total LAPNs labeled. All analyses were performed by experimenters blinded to the experimental conditions. Image processing and a priori inclusion/exclusion criteria for analyses are previously described (*Pocratsky et al., 2017*).

Laminar distribution analyses and heatmap generation were performed as previously described (*Pocratsky et al., 2017*). To generate contour plots, neurons were first marked using Nikon Elements software. A custom-made MatLab program was then developed to reconstruct and normalize the position of labeled neurons across sections. A reference axis was created for each image with the origin centered on the central canal, the y-axis parallel to the spinal cord midline, and the x-axis orthogonal to the y-axis (*States, 2020*; copy archived at https://github.com/elifesciences-publications/Pocratsky_et_al_2020). Contour/scatter plotting was performed using R. Distribution contours were created by calculating the two-dimensional kernel density (using the kde2d function in the MASS library), then connecting points of equal density values between 30–100% of the estimated density range in increments of 10% (*States, 2020*).

Immunohistochemical detection of putative synaptic inputs onto LAPNs was performed in accordance with methods previously described (*Pocratsky et al., 2017*).

## Statistical analyses

Statistical analyses were performed using the SPSS v22 software package from IBM. Additional references for parametric and non-parametric testing were used in complementation to SPSS (*Hays, 1981*; *Siegel and Castellan, 1988*; *Batschelet, 1972*; *Zar, 1974*; *Ott, 1977*; *Lenth, 2006*). Differences between groups were deemed statistically significant at $p \leq 0.05$. Two-tail p values are reported.

The Binomial Proportion Test was used to detect significant differences in the proportion of coordination values beyond control threshold for the raw and transformed interlimb coordination data of various limb pairs. It was also used to detect a significant group peak effect (Dox1 vs Dox2), per-step changes in left-right coordination and stride durations (beyond control thresholds), the interaction between altered coupling patterns, testing for the preferred "altered" forelimb coupling pattern during silencing, the stroke-by-stroke changes in hindlimb coordination as well as stroke cycle durations (beyond control variability), and the various behavioral contexts (e.g. stepping surface).

Circular statistics were performed on the stepping inter- and intralimb coordination datasets, as well as the swimming hindlimb coordination data (*Pocratsky et al., 2017*; *Zar, 1974*). We primarily used the non-parametric two-sample U (*Orlovskiĭ et al., 1999*) test for the following rationale. Typically, parametric tests are performed to determine whether the data have a uniform distribution (*Batschelet, 1972*; *Zar, 1974*). Importantly, these analyses are based on strict assumptions that the distribution is restricted to two patterns: uniform or unimodal (*Batschelet, 1972*; *Zar, 1974*). Our data do not fit these criteria (e.g. differences in lead limb and natural intra- and inter-animal variability in interlimb coordination). Moreover, the various control time points (Baseline, Pre-Dox1, Dox$^{Off}$, Pre-Dox2) do not have unimodal distributions with the exact same degree of concentration. Therefore, we used non-parametric two-sample U (*Orlovskiĭ et al., 1999*) test. The null hypothesis tested here is whether two time points have the same concentration (or phasic direction) in couple pattern expression.

Spearman Rank correlations were performed on the speed versus spatiotemporal gait indices for the forelimbs and hindlimbs during Control and Dox, respectively. These comparisons included speed versus stance, swing, and stride durations as well as the stride length and frequency.

Regression analyses to compare the slopes for the lines of best fit were performed on the speed versus spatiotemporal gait indices datasets (Control vs Dox for forelimbs and hindlimbs, respectively, as well as between the limb pairs). Regression and slope analyses were also performed to test for preferred coupling patterns in the altered stepping datasets as well as comparing the left versus right fore- and hindlimb step frequency and durations as well as comparing between the two girdles.

One and two-way ANOVAs were used to test for significant differences in the laminar distribution and projection patterns of LAPNs as previously described (*Pocratsky et al., 2017*).

Mixed model ANOVA followed by Bonferroni post hoc t-tests (where appropriate) were used to detect a significant difference in the peak, trough, and excursion of the proximal and distal hindlimb segments for range-of-motion analyses.

Repeated measures ANOVA without speed as a co-variate were performed when comparing the mean stride durations between the fore- and hindlimbs within the individual time points.

Repeated measures ANOVA with speed as a co-variate were used when comparing Control vs Dox stride, swing, and stance durations for the fore- and hindlimbs as well as between the girdles. Sidák post hoc t-tests were used when appropriate.

Multivariate analysis of variance (MANOVA) with speed as a co-variate followed by Sidák post hoc t-tests were used when comparing the mean stride frequencies and durations for Control vs Dox for the fore- and hindlimbs as well as between the two girdles. These analyses were also used when comparing the average stride durations of the left and right forelimbs and hindlimbs, respectively, over time (nine total time points, excluding vehicle control) as well as within the individual time points.

Paired t-tests were used to detect significant differences in: (1) the magnitude change in interlimb coordination during silencing, (2) the proportion of steps with per-stride changes that were $\leq 0.1$ or $> 0.1$, (3) the hindlimb:forelimb step index, (3) when comparing the percent of Dox steps that were

≤90 cm/s versus >90 cm/s as well as (4) for the altered steps alone, (5) when comparing the base-of-support, (6) average number of foots slips on the ladder (7) and beam (8 , 9) the frequency and (10) duration of spontaneously expressed rearing events, (11) the trunk angle during swimming, (12) when comparing the swing-stance durations within speed categories of ≤90 cm/s or >90 cm/s for the fore- and hindlimbs, respectively, at Control and Dox, and comparing the coefficient of variation at Control and Dox time points.

Levene's Test for Equality of Variances were performed to test for a normal distribution within the interlimb coordination datasets. Notably, at control time points (e.g. Baseline) the coordination data have a non-normal distribution as phase values will naturally concentrate towards one value (e.g. 0.5 for left-right alternation in the hindlimbs).

## Code availability

Kinematic and gait data were analyzed using custom-built Excel add-in macros (*Morehouse, 2020*; copy archived at https://github.com/elifesciences-publications/KSCIRC-Gait-Addin). Heatmaps and contour plots of LAPN laminar distribution were generated using custom-designed MatLab and R scripts (*States, 2020*; copy archived at https://github.com/elifesciences-publications/Pocratsky_et_al_2020).

## Acknowledgements

The authors thank Dr. Tadashi Isa and Dr. Akiya Watakabe for providing the silencing viral vector plasmids, Russell M Howard for assistance in vector production, Christine Yarberry for surgical support and Alice Shum-Siu for histological assistance.

## Additional information

### Funding

| Funder | Grant reference number | Author |
| --- | --- | --- |
| National Institute of Neurological Disorders and Stroke | R01 NS089324 | Scott R Whittemore<br>David SK Magnuson |
| National Institute of Neurological Disorders and Stroke | P30 GM103507 | Scott R Whittemore<br>David SK Magnuson |
| Kentucky Spinal Cord and Head Injury Research Trust | 13-14 | Scott R Whittemore<br>David SK Magnuson |

The funders had no role in study design, data collection and interpretation, or the decision to submit the work for publication.

### Author contributions

Amanda M Pocratsky, Conceptualization, Data curation, Formal analysis, Investigation, Methodology, Project administration; Courtney T Shepard, Formal analysis, Investigation; Johnny R Morehouse, Formal analysis, Validation, Methodology; Darlene A Burke, Data curation, Formal analysis, Validation, Visualization; Amberley S Riegler, Gregory JR States, Formal analysis, Investigation, Visualization; Josiah T Hardin, Casey Hainline, Data curation, Formal analysis, Visualization; Jason E Beare, Formal analysis, Visualization; Brandon L Brown, Formal analysis, Methodology, Writing - review and editing; Scott R Whittemore, Conceptualization, Supervision, Funding acquisition, Validation, Project administration; David SK Magnuson, Conceptualization, Supervision, Funding acquisition, Validation, Visualization

### Author ORCIDs

Jason E Beare ⓘD http://orcid.org/0000-0003-3988-1223
Scott R Whittemore ⓘD https://orcid.org/0000-0001-6437-7200
David SK Magnuson ⓘD https://orcid.org/0000-0003-3816-3676

### Ethics

Animal experimentation: This study was performed in strict accordance with the recommendations in the Guide for the Care and Use of Laboratory Animals of the National Institutes of Health. All of the animals were handled according to the approved institutional animal care and use committee (IACUC) protocol (#16669) of the University of Louisville. All surgery was performed under sodium pentobarbital or isoflurane anesthesia, and every effort was made to minimize suffering.

### Decision letter and Author response

Decision letter https://doi.org/10.7554/eLife.53565.sa1
Author response https://doi.org/10.7554/eLife.53565.sa2

## Additional files

### Supplementary files

• Supplementary file 1. The fundamental relationship between speed and the various speed-dependent locomotor parameters remains intact despite the silencing-induced disruption to interlimb coordination. Time point comparisons reveal that silencing long ascending propriospinal neurons (LAPNs) did not affect the underlying foundation relationship between speed and stance duration, stride duration, and stride length at the fore- and hindlimbs, respectively. Running the same comparisons on the curated dataset of Dox-induced affected stepping yielded similar results ('Dox-induced affected step'). Pearson correlation coefficients (r) and p values shown post-Bonferroni correction for multiple comparisons.

• Supplementary file 2. Silencing long ascending propriospinal neurons (LAPNs) functionally uncouples the intra-girdle left-right limb pairs during overground locomotion. Interlimb coordination data were analyzed using Watson's non-parametric two-sample U (*Orlovskiĭ et al., 1999*) test (Critical value of Watson's $U^2$ = 0.1869; Appendix D, TableD.44) (*Zar, 1974*)

• Supplementary file 3. Silencing long ascending propriospinal neurons (LAPNs) disrupts interlimb coordination in select behavioral contexts. Interlimb coordination data were analyzed using Watson's non-parametric two-sample U (*Orlovskiĭ et al., 1999*) test (Critical value of Watson's $U^2$ = 0.1869; Appendix D, TableD.44) (*Zar, 1974*)

• Transparent reporting form

### Data availability

Source data has been provided for: Figures 2, 3, 4 and 5, Figure 1—figure supplement 1 and Figure 4—figure supplement 2.

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
