## [Decision Letter]

**Acceptance summary:**

Procratsky and colleagues provide an anatomical and functional evaluation of long ascending propriospinal neurons (LAPNs) connecting lumbar hindlimb-related segments and cervical forelimb-related segments. They use an elegant method (developed by Tadashi Isa) to specifically and reversibly silence LAPNs in rat, expecting to uncouple fore/hindlimb coordination. Instead, they observe changes in left-right coupling that occur only during non-exploratory locomotion on high friction surfaces not for example during swimming, treadmill stepping, or on a slick surface.

**Decision letter after peer review:**

Thank you for submitting your article "Long ascending propriospinal neurons provide flexible, context-specific control of interlimb coordination" for consideration by *eLife*. Your article has been reviewed by Ronald Calabrese as the Senior Editor, a Reviewing Editor, and three reviewers. The following individuals involved in review of your submission have agreed to reveal their identity: Tuan V Bui (Reviewer #3).

The reviewers have discussed the reviews with one another and the Reviewing Editor has drafted this decision to help you prepare a revised submission.

Summary:

This paper explores the functional role of long-range ascending projection neurons (LAPNs) in rat spinal cord. Procratsky and colleagues provide an anatomical and functional evaluation of LAPNs, which connect lumbar hindlimb-related segments and cervical forelimb-related segments. They use an elegant method (developed by Tadashi Isa) to specifically and reversibly silence LAPNs – doxycycline-mediated TeNT silencing of LAPNs whose cell bodies reside mainly in L1-L3 segments – expecting to uncouple fore/hindlimb coordination. Instead, they observe changes in left-right coupling that occur during non-exploratory locomotion on high friction surfaces, but not during treadmill locomotion, nose-down exploratory locomotion, slippery surface locomotion and swimming. Overall, the experiments are done rigorously. Several controls have been done to suggest that neurons affected by Dox are indeed long ascending propriospinal neurons from L1-L3. Many features of locomotor activity have been taken into consideration in their analysis, and data capture for locomotor activity used several different approaches and appears well done. The study of different forms of locomotor activity provides greater insight into the possible roles of long ascending propriospinal neurons and is a strength of this work. The insights from this study advance our understanding of how spinal circuitry regulates different facets of locomotor activity and pushes the field to consider that the contributions of spinal circuits shift depending on the type of activity performed.

Essential revisions:

All the reviewers were supportive of the work, while voicing significant concerns. All the complementary concerns of the reviewers should be addressed. Two issues arose as shared concerns of the reviewers.

1) We require further analysis of which neurons the viral strategy is labeling/manipulating, perhaps using the tissue used in Figure 2. Is this labeling similar to the CtB? If not, how? Are the differences laminar-related? Ideally, the authors should show that most or a subset of CtB labeled neurons are eTeNT.GFP+, and more importantly that >95% of eTeNT.GFP+ are CtB+. Minimally, there should be comparative analysis of the maps of each.

2) We require a thorough analysis of speed in all the forms of locomotion tested. The potentially most exciting aspect of the work is the context dependence. However, the authors must demonstrate that the other behaviors that were unperturbed were operating over the same speed range. Otherwise this changes the interpretation altogether. Instead of the same movement pattern being perturbed in one context but not in the other, it would instead suggest that a particular movement pattern is disrupted, and this is most obvious in contexts when that pattern is utilized.

Reviewer #1:

This paper explores the functional role of long-range ascending projection neurons (LAPNs) that connect rostral lumbar and mid-caudal cervical spinal enlargements in rats. The authors use a viral combinatorial strategy that can selectively and reversibly silence neurons based on their projection patterns. The authors then use a battery of behavioral tests to reveal the impact of LAPNs on coordination within and between the limbs. While no impact was observed in coordination within limbs, the authors find that coordination between limbs was disrupted by their perturbations. Strikingly, this effect was observed not only for coordination between hindlimbs and forelimbs (as expected based on their projection patterns), but also for coordination between the left- and right sides. This was thought to rely on local connections and now appears to also rely on these LAPNs. Since the projection patterns are the only defining feature presented here, the identity of these neurons is still unclear. However, the authors demonstrate that this impact is context-dependent, meaning that locomotion in different behavioral contexts is not always consistently altered. Specifically, the phenotype is most obvious during overground locomotion on a gripping surface, but not during treadmill locomotion, nose-down exploratory locomotion, slippy surface locomotion and swimming. The writing is clear and the figures are beautiful, however I have some suggestions regarding analysis and interpretation that I hope help bolster their conclusions.

Major comments:

1) The authors state that chemical based tracers can label fibers of passage, while viruses do not (subsection “Histological detection of putatively silenced LAPNs”). It seems critical to confirm that their viral labeling is labeling the same populations in L1-3 as their tracers do. There are examples provided for labeling in Figure 2J-L, but no detailed segmental or laminar analysis as provided for tracer labeling. It is also not clear what sort of variability from animal to animal they observed in labeling. Was it always bilateral and limited to laminae 6-7 in L1-3? Without a better idea of which neurons were labeled where and how reliably, it is difficult to interpret the subsequent behavioral tests. For example, subsection “LAPNs organize interlimb coupling at each girdle during overground stepping”, were different stepping behaviors observed within the same animal or different gaits in different animals? Could differences in extent of labeling account for weaker versus stronger effects?

2) As I understand it, eTeNT and GFP expression should be linked and activated by retrograde transport to the soma. So, it's not clear to me why somatic GFP labeling would be much dimmer than axon terminal labeling, if eTeNT and GFP at the terminals are arriving anterogradely from the soma (at least that is what I surmised from the need for GFP signal amplification in subsection “Histological detection of putatively silenced LAPNs”). Apologies if I've misunderstood something. A more systematic analysis of eTeNT-GFP expression patterns along the rostrocaudal axis would help with this concern too.

3) From the contour density analysis in Figure 1H-J, there appear to be differences in the relative bilateral distribution of ipsi and contra cells (if I am interpreting the yellow and blue lines correctly) that indicate systematic differences in the distribution of ipsi versus contra LAPNs as you move caudally. Also, there are differences in lamina distribution in L3 compared to L1 and L2 (Figure 1M-O). It is difficult to understand the functional implications or why the authors carried out this analysis without a bit more information. All of the data are normalized to total, so it is difficult to get a handle on real numbers and variation.

4) The lack of any sort of identification, either by transcription factor or by transmitter phenotype, makes it difficult to generalize to other locomotor networks. Although glutamatergic and GABAergic axon terminals are identified, the source (whether ipsilateral or contralateral, L1,2,3 or elsewhere) is still unclear. Molecularly-defined excitatory and inhibitory spinal interneurons can migrate some distance from their point of origin, but tend to settle in consistent regions. If laminar distribution is an important clue to their identity (sensory, motor, other), it should be more clearly stated.

5) Rhythms within a limb aren't effected so I think it's safe to say that these LAPNs are not rhythm-generating. However, without a better idea of the identity of these neurons, one cannot rule out the possibility that they are sensory interneurons relaying proprioceptive or exteroceptive signals. I think this possibility should be raised in the Discussion section along with a potential pattern-forming motor function.

6) Since different speeds of locomotion are used in different behavioral contexts, it is difficult to separate which of these two features is more important with the current analysis. For example, if bilateral synchrony is observed at faster speeds, then behaviors that are slow would not be affected. It would be worth plotting the phase data as a function of speed in the overground behavioral tasks (e.g., Figure 3), to see if bilateral activity becomes more obvious at faster speeds or if it is observed over the entire speed range. Similarly, it would be good to know the speed range/cycle duration of the other tasks (e.g., treadmill stepping subsection “Silencing LAPNs disrupts interlimb coordination independent from the salient features of locomotion.”) to see how they may overlap. For example, the Arber lab observed no effects on slow treadmill locomotion when they ablated LDPNs, only at fast speeds was a deficit observed. I couldn't find the range of speeds used for treadmill locomotion, but these should be reported.

7) From Figure 5, it looks like the Dox treated mice are capable of moving faster (cm/s) than controls (panels C-F, H-K, M-P). Could this also explain the increased co-contraction? They are operating in a higher speed regime for more time? Plotting phase against frequency in control and Dox treated animals would also help determine whether this is a context- versus speed-dependent phenomenon.

Reviewer #2:

Procratsky and colleagues provide an anatomical and functional evaluation of long ascending propriospinal neurons (LAPNs) connecting lumbar hindlimb-related segments and cervical forelimb-related segments. They use an elegant method (developed by Tadashi Isa) to specifically and reversibly silence LAPNs in rat, expecting to uncouple fore/hindlimb coordination. Instead, they observe changes in left-right coupling that occur only during non-exploratory locomotion on high friction surfaces. This study is well-presented and the datasets are comprehensive, and the findings are thought-provoking.

In addition to the unexpected (and quite puzzling) primary result, there are other key findings of interest, including the identification of spinal neurons involved in locomotion in a context-dependent manner, and a potential demonstration that disruption of locomotor pattern that does not affect the rhythm. Further, it is exciting to see reversable silencing experiments in an animal model aside from transgenic mice. These results are novel and contribute to our understanding of the spinal locomotor circuit.

Major comments:

1) For the interpretation of manipulation studies, it is essential to know which neurons are being targeted. The CtB and eTeNT.EGFP histology presented does not directly get at this issue. On one hand, the CtB data provides a thorough description of lumbar LAPNs. However, the overlap with the neurons being manipulated with the eTeNT is missing and, as pointed out by the authors, CtB labels fibers of passage which belong to neurons that are not being silenced. This raises the question of whether the eTeNT.GFP directly overlaps with the CtB-labeled populations, just the numbers are less, or are there specific regions where there are eTeNT.GFP neurons within the more widespread CtB labeling? This is not possible to determine with the results presented in Figure 2. A mapping similar to what was performed for the CtB data would allow the reader to compare and would be helpful to assess exactly what is being manipulated.

2) I'm not entirely convinced one can conclude that the local commissural projections of the LAPNs are minor from the data presented. According to Figure 1—figure supplement 1, it is about 10% of the commissural LAPNs that do have projections to L1 and L5 (30% of 16% in L1 + 45% of 9% in L5 = ~10%). This is not counting any that may project within segments L2-L4 and even if that's just an additional 5%/segment, that could be 25% which is substantial. The tracer would have needed to be injected into a wider region of the contralateral cord and a low overlap observed to make this conclusion (but if the experiment were to be performed in this way, information regarding ipsilateral LAPNs would have been lost).

3) Following the previous point, there may be significant overlap with the V0 and V2a populations here. It is impossible to know for certain as the overlap of the LAPNs manipulated here with genetic populations cannot be determined and the mouse work does not detail the degree to which local vs LPNs are manipulated. Where the presented locomotor phenotype is similar in some ways to the phenotype seen in V0V and V2a mutants (i.e. left/right synchrony are more prominent and observed at a lower locomotor frequency ), there are distinct differences between the findings here and the mouse studies (trot is lost in the mouse mutants but it is present in the rats, the speed profiles are more compressed in the genetic mutants and that does not seem to be the case here, LAPN silencing effects depend on condition, etc.). Can these experiments be considered as complementary and, if so, does this provide additional insight into the circuitry?

4) It seems that the figure that was uploaded as Supplementary Figure 5 is incorrect. The figure is identical to Figure 5 but figure legend does not match.

5) In the Discussion section, although it was clear that the authors expected that forelimb and hindlimb coordination would be disrupted and the lack of that finding is well-described, there is no discussion about why this is not the case. Additionally, the suggestion of LAPNs being "distributors of temporal information" comes up a few times (Discussion section). How that may result in the locomotor changes seen in the data is not obvious and it would help to have an expansion of that idea for clarification.

Reviewer #3:

Long propriospinal neurons link motor circuits involved in the control of forelimbs and hindlimbs. Locomotion requires the constant coordination of forelimb and hindlimbs and this likely involves long propriospinal neurons. To what extent these neurons influence the rhythm and pattern of limb activity across limbs and within each girdle is not well understood. This study uses doxycycline-mediated TeNT silencing of long ascending propriospinal neurons whose cell bodies reside mainly in L1-L3 segments of the spinal cord to provide insights into the role of long propriospinal neurons in limb coordination. Their results suggest that these neurons are involved with intra-girdle and interlimb coordination in a manner that is independent of the regulation of general features of locomotion such as limb kinematics, as well as stride times and lengths, and durations of different phases. Interestingly, the disruption of coordination occurs within only certain types of activity such as overground stepping, and not during treadmill walking or exploratory activity.

Overall, the experiments are done rigorously. A number of controls have been done to suggest that neurons affected by Dox are indeed long ascending propriospinal neurons from L1-L3. Many features of locomotor activity have been taken into consideration in their analysis. Data capture for locomotor activity used several different approaches and appears well done, however, certain experimental details need some clarification (see discussion of treadmill experiments below). The study of different forms of locomotor activity provides greater insight into the possible roles of long ascending propriospinal neurons and is a strength of this work. The insights from this study advance our understanding of how spinal circuitry regulates different facets of locomotor activity and pushes the field to consider that the contributions of spinal circuits shift depending on the type of activity performed.

We have one substantive concern to raise that is related to whether there is a speed-dependence related to the loss of coordination. This issue stems from data illustrated in Figure 5 where steps in Dox-treated animals are colour-coded to show the steps that are unaffected (red) and those that are affected (blue). It looks like affected steps (blue) are on average occurring at higher speeds than those that are not (red). This suggests that loss of coordination is more likely at higher speeds in Dox-treated animals. Could it be that the differences seen in a different context are purely related to the speed of locomotion achieved in these different locomotor activities? The Dox-treated animals appear to have a greater proportion of strides performed at a fast pace (>90 cm/s) than controls in the volitional locomotion experiment (Figure 5C-P). Perhaps re-analyzing using a sample of steps uniformly distributed across speeds in control and Dox-treated animals could answer whether loss of coordination is speed-dependent or context-dependent.

In the same line as above, we did not find details related to the speeds tested for the treadmill experiments. Were the animals tested on the treadmill at multiple speeds (ie. slow, medium, fast)? If so – what speeds were tested and were any differences noted? Similarly, were the average walking speeds different between trials on uncoated and Sylgard-coated surfaces? These details could provide a common factor that explains the apparent context-dependence of the data.

Finally, while the data clearly demonstrate abnormalities in intra-girdle patterning following inhibition of the LAPNs, the basis for these abnormalities isn't clear. The effects on the hindlimb pattern are particularly perplexing if we assume that since the LAPNs are unlikely to directly interact with the lumbar CPG based upon their sparser connectivity within the lumbar spinal cord. Rather than the LAPNs being necessary for intra-girdle coordination, is it not more plausible that they act by compensating for the proprioceptive confound caused by forward propulsion from the opposite girdle? This could also help to explain why volitional locomotion – where there is likely more variation in speed (and thus more sensory confounds from acceleration and deceleration) – exhibited more gait abnormalities than treadmill walking. While this difference in interpretation might seem trivial, it would help to fit these results within the context of spinal cord injury literature – wherein loss of LAPN connections to the cervical spinal cord by thoracic transection has not (to our knowledge) produced similar forelimb gait abnormalities. In these spinal cord injured animals, the hindlimbs do not produce forward propulsion, and therefore do not have a destabilizing influence on forelimb proprioception/CPG function – thus not necessitating LAPN input to maintain appropriate forelimb alternation. Further discussion or justification for the current interpretation of the results should be provided.

[Editors' note: further revisions were suggested prior to acceptance, as described below.]

Thank you for resubmitting your article "Long ascending propriospinal neurons provide flexible, context-specific control of interlimb coordination" for consideration by *eLife*. Your revised article has been reviewed by Ronald Calabrese as the Senior Editor, a Reviewing Editor, and three reviewers. The following individuals involved in review of your submission have agreed to reveal their identity: Tuan V Bui (Reviewer #3).

The reviewers have discussed the reviews with one another and the Reviewing Editor has drafted this decision to help you prepare a revised submission. In recognition of the fact that revisions may take longer than the two months we typically allow, until the research enterprise restarts in full, we will give authors as much time as they need to submit revised manuscripts.

Summary:

The paper has been much improved by new experiments, clarifications, and rewriting. Nevertheless, significant changes are needed before the paper can be published in *eLife*.

Essential revisions:

Quantification (alternative changes provided):

In response to our comments the authors have provided a viral labeling method that better matches the approach used to silence neurons. While it is still not the same construct, it is a closer approximation. Indeed, this demonstrates that the method used to characterize LAPNs in Figure 1 overestimates the populations labeled. However, the authors have chosen to focus exclusively on ipsilateral populations, when data from Figure 1 clearly illustrate that commissural populations make up a larger fraction of the LAPN population (significant difference, little overlap in values, but downplayed in the main test). The new data are included as 'supplemental', when these are the key experiment to demonstrate which populations are labeled. Since much of the analysis in Figure 1 is not raised in the results and since the authors have already published data describing the populations of LAPNs in the lumbar spinal cord, this figure no longer seems relevant. Instead we suggest one of the following:

Alternative 1

a) A more detailed analysis of the new viral labeling strategies and their distribution on ipsilateral and contralateral sides of L1-2 to start out as Figure 1. This would provide more space to outline the caveats of the approach to the reader up front.

b) Move current Figure 1 into supplemental and provide some context and interpretation to the analysis you have performed and how it differs from their previous study (Reed et al., 2006). The authors say in their response that they do not want to speculate, but at least accurately reporting differences in distribution where they exist (e.g., L3 versus L1-2) is warranted. Particularly since labeling in L3 looks more like their new viral data. Also, please use bold colors instead of a gradient to mark laminae, since this is challenging to discern distinct laminae based on subtle shade differences. For this figure, it would also be more helpful for ease of interpretation, if the contour density analysis was pooled so that contra and ipsi populations are on different sides, not as they are based on dye labeling approaches (where one gets the impression one dye is better at labeling than the other or that contra populations also have ipsi projections or both).

Alternative 2

a) Remove current Figure 1 altogether and replace with more detailed analysis of the new viral labeling data.

Specificity of labeling (alternative changes provided):

The bright local labeling presented in Figure 2 is explained in the response to reviews, but not in the main text. The utility of this figure is also not clear, since the new labeling data better serve this purpose. Also, the new data suggest that axons of passage can be labeled by their CTB label, yet there are no L2 neurons labeled by injecting in L1 (Figure 1—figure supplement 1). Some clarification here is required, since the current argument is that spinocerebellar neurons are labeled as axons of passage. The point of this experiment is also not clearly articulated and just adds confusion.

We suggest one of the following:

Alternative 1:

a) Moving Figure 2 to supplemental, with a stronger statement or data supporting the idea that this does not reflect more proximal ectopic labeling. For example, were sections performed in cervical or thoracic levels to rule this out? Is this technically impossible or controlled for in past work? In addition, marking the laminae in which the labeled neurons are located would help link to new viral labeling work.

b) Provide a better justification for the experiments and an interpretation of the lack of labeling using local injections.

Alternative 2:

a) Removing Figure 2, since the new labeling experiments serve this purpose and do not raise as many questions.

Data curation (alternative changes provided):

Much of the critical data is currently found in supplemental, which could replace current main figure components that convey the same information. For example, in Figure 3, the running mouse illustrations are beautiful in panels c-e, but they take up a large fraction of the figure and add little beyond what is presented in panel b with respect to differences in coordination. Also, Figure 3 sets up the interlimb coordination differences, then moves to Figure 4 which demonstrates no impact on intralimb effects, then moves back to interlimb differences in Figure 5, requiring another cartoon summary of different gaits. Space could be saved for the inclusion of supplemental data if the figures are re-ordered. We suggest one of the following:

Alternative 1:

a) Re-order the figures, so that a new Figure 1 deals with the method of labeling and distribution of neurons with bilateral analysis, Figure 2 deals with the lack of impact on rhythm generation, as expected. Then Figure 3 and Figure 4 move into the impact on left-right alternation and limb coordination. In particular, moving data from Figure 5—figure supplement 2 into the main figures is strongly encouraged.

Alternative 2:

a) Keep figure ordered as they are, but reduce redundant information in figures (e.g., introductory cartoons) to make room for supplemental data that make key arguments (e.g., Figure 5—figure supplement 2). For example, panel 3C could be incorporated into panel 3B and panels 3B and 5A essentially convey the same information.

Speed versus context control:

The revision still does not rule out the possibility that Dox silencing is simply most obvious in contexts when the animal must move at faster speeds. It seems clear that silencing pushes the animals into a faster range of frequencies and the premature adoption of gaits normally used during very fast locomotion. This would be consistent with the impact of disruptions to spinal interneuronal control circuits in mammals and bolster their argument that LAPNs play similar roles in coordination. For over-ground locomotion (Figure 5—figure supplement 2), altered steps dominate the shift to higher speeds (cm/s). For treadmill locomotion, the mouse does not reach the higher range of speeds observed during over-ground locomotion and instead appears to use a strategy more reliant on longer stride times and lower stride frequencies. So, the lack of altered steps could be because the animal is not pushed to move as fast and adopts a different locomotor strategy (that is still impacted by Dox silencing, but not as much because deficits are always more obvious at faster speeds). The same argument could be made for exploratory mode walking, where the animal is not pushed to move as fast. The coated surface is like over-ground because they can reach faster speeds, while the smooth one is a bit slower (cm/s wise) and so not as impacted. Overall, it seems that deficits evoked by Dox silencing are most obvious in contexts in which the animal needs to move faster, not because they are moving the same way in different contexts however Dox silencing preferentially impacts one context. To better rule out this possibility, we suggest the following:

a) Statistical tests of the overall speeds between tasks in addition to the within-tasks analysis of altered versus unaltered that you've already performed. This is presented in Figure 5—figure supplement 2E, but no stats are reported, and it looks like the movements most greatly impacted are ones that have the highest range of frequencies (over-ground and coated, grey bars). If this is significant, the idea that speed is playing a role needs to be considered in the interpretation presented.

---

## [Author Response]

Essential revisions:All the reviewers were supportive of the work, while voicing significant concerns. All the complementary concerns of the reviewers should be addressed. Two issues arose as shared concerns of the reviewers.1) We require further analysis of which neurons the viral strategy is labeling/manipulating, perhaps using the tissue used in Figure 2. Is this labeling similar to the CtB? If not, how? Are the differences laminar-related? Ideally, the authors should show that most or a subset of CtB labeled neurons are eTeNT.GFP+, and more importantly that >95% of eTeNT.GFP+ are CtB+. Minimally, there should be comparative analysis of the maps of each.

We did not have sufficient tissue from the animals that were assessed behaviorally to perform the exact comparison requested. However, we generated an independent set of animals using viral tracing and detailed both the numbers of infected cells and their laminar location for unilaterally projecting LAPNs. The new supplemental figure 2 quantifies those data and shows heat maps and neuron counts that are compared directly with the existing heat maps of CTB^+^ neurons. We observe that the comparison between the CTB and virally-labeled neurons highly corresponds with the outlier being the larger percentage of CTB^+^ lamina 5 neurons. These likely reflect spinocerebellar neurons (Matsushita, 1999; Arsenio Nunes and Sotelo, 1985) whose fibers of passage were labeled by CTB. These data highlight the greater specificity of viral labeling which is based both on cell soma and axon terminal locations. These results and relevant discussion have been added to the text.

2) We require a thorough analysis of speed in all the forms of locomotion tested. The potentially most exciting aspect of the work is the context dependence. However, the authors must demonstrate that the other behaviors that were unperturbed were operating over the same speed range. Otherwise this changes the interpretation altogether. Instead of the same movement pattern being perturbed in one context but not in the other, it would instead suggest that a particular movement pattern is disrupted, and this is most obvious in contexts when that pattern is utilized.

We appreciate this critical feedback and agree that a more in-depth exploration of speed is necessary across the behavioral contexts. Our new Figure 5—figure supplement 2 focuses on the speed ranges observed across the various behavioral conditions. We concentrated on the hindlimbs as this was the site most affected by LAPN silencing and presented the data in a format similar to that of Figure 5 (speed vs hindlimb stride time/frequency/length) as it clearly illustrates the full variability observed across all datasets.

Key outcomes illustrated include:

a) The speed range (minimum-maximum) observed across each behavioral condition at Control time points (gray inset in panels a-dd). These values reflect the instantaneous, per-step speed observed with each successive step cycle. Outcome(s): the various behavioral conditions show overlapping speed ranges at Control time points.

b) Expanded speed ranges observed during Dox^On^ are illustrated with pink insets. Outcome(s): apart from treadmill stepping, the speed range observed is expanded during Dox^On^ LAPN silencing (detect slower and faster speeds as compared to control time points). The increased expansion observed is still well below the speed at which synchronous-like gaits such as half and full bound are observed in adult rats (as shown in Figure 5 and previously published – references previously reported within text).

c) The Dox^On^ data (with irregular steps shown in yellow) is overlaid onto the control speed range. Outcome(s): across each behavioral condition, the majority of Dox^On^ steps fall within speed ranges observed at control time points.

d) The percent of silencing-induced irregular steps that fall within the control speed range across is behavioral context is reported in panels b, d, f, h, and j. Outcome(s): the preponderance of altered step cycles falls within the control speed range for each behavioral condition.

e) All speed ranges observed across each behavioral context are plotted together in panel ee. Data reported includes: (i) the speed range observed at control time points (gray bars), (ii) Dox^On^ time points (red bars), and (iii) Dox^On^-induced irregular steps (yellow bars). (iv) Control and Dox^On^ averages and standard deviations are overlaid onto the corresponding bar plots.

f) The overground stepping data was binned for further analysis where we compared the proportion of steps that were ≤90 cm/s vs >90 cm/s for all Dox^On^ steps as well as the irregular steps only. Analyses were performed for left-right forelimb and left-right hindlimb coordination, respectively. Outcome(s): the majority of DoxOn steps were ≤90 cm/s and we saw no association between higher speeds (>90 cm/s) and number of irregular steps observed (hypothesis tested: if speed was a contributing factor, then we would see a significant proportion of irregular steps taken at >90 cm/s).

g) data not shown but reported in results (subsection “Silencing-induced disruption to interlimb coordination is context-dependent”). The phase-speed relationship was also quantitatively assessed, testing the hypothesis that changes in speed correlate with changes in coordination (ergo gait switches). We first tested this association in our long-tank gait dataset where rats expressed their full gait repertoire at the corresponding naturalistic speeds. As expected, we saw a predictable relationship between increased speed and corresponding changes to hindlimb coordination (Spearman Rank correlation coefficient = 0.753, N=12 age-matched control rats, n=403 total steps analyzed). When we ran similar comparisons for the Dox^On^ overground stepping dataset, no predictable relationship was detected with a weak correlation coefficient of 0.410 (N=13 rats, n=600 total steps analyzed).

These additional analyses show that animals (with or without silencing) express overlapping speed ranges across the various behavioral contexts. Silencing LAPNs does expand the speed range expressed. The expansion of speeds observed may be a consequence, but not a cause of the totality of silencing-induced changes to interlimb coordination.

Reviewer #1:This paper explores the functional role of long-range ascending projection neurons (LAPNs) that connect rostral lumbar and mid-caudal cervical spinal enlargements in rats. The authors use a viral combinatorial strategy that can selectively and reversibly silence neurons based on their projection patterns. The authors then use a battery of behavioral tests to reveal the impact of LAPNs on coordination within and between the limbs. While no impact was observed in coordination within limbs, the authors find that coordination between limbs was disrupted by their perturbations. Strikingly, this effect was observed not only for coordination between hindlimbs and forelimbs (as expected based on their projection patterns), but also for coordination between the left- and right sides. This was thought to rely on local connections and now appears to also rely on these LAPNs. Since the projection patterns are the only defining feature presented here, the identity of these neurons is still unclear. However, the authors demonstrate that this impact is context-dependent, meaning that locomotion in different behavioral contexts is not always consistently altered. Specifically, the phenotype is most obvious during overground locomotion on a gripping surface, but not during treadmill locomotion, nose-down exploratory locomotion, slippy surface locomotion and swimming. The writing is clear and the figures are beautiful, however I have some suggestions regarding analysis and interpretation that I hope help bolster their conclusions.Major comments:1) The authors state that chemical based tracers can label fibers of passage, while viruses do not (subsection “Histological detection of putatively silenced LAPNs”). It seems critical to confirm that their viral labeling is labeling the same populations in L1-3 as their tracers do. There are examples provided for labeling in Figure 2J-L, but no detailed segmental or laminar analysis as provided for tracer labeling. It is also not clear what sort of variability from animal to animal they observed in labeling. Was it always bilateral and limited to laminae 6-7 in L1-3? Without a better idea of which neurons were labeled where and how reliably, it is difficult to interpret the subsequent behavioral tests. For example, subsection “LAPNs organize interlimb coupling at each girdle during overground stepping”, were different stepping behaviors observed within the same animal or different gaits in different animals? Could differences in extent of labeling account for weaker versus stronger effects?

See comment 1 above and the new Figure 5—figure supplement 2.

2) As I understand it, eTeNT and GFP expression should be linked and activated by retrograde transport to the soma. So, it's not clear to me why somatic GFP labeling would be much dimmer than axon terminal labeling, if eTeNT and GFP at the terminals are arriving anterogradely from the soma (at least that is what I surmised from the need for GFP signal amplification in subsection “Histological detection of putatively silenced LAPNs”). Apologies if I've misunderstood something. A more systematic analysis of eTeNT-GFP expression patterns along the rostrocaudal axis would help with this concern too.

The reviewer is correct – the transgene expressed is an eTeNT-EGFP fusion protein which is expressed in double infected neurons, but is only activated when doxcycyline is provided ad libitum. The dimmer labeling in the somata has puzzled us as well. We’ve attempted to address this issue by adjusting various technical parameters to no avail, including changes to the fixation (e.g. overnight vs 1-3 hours post-fixation) with or without antigen retrieval, using various anti-GFP antibodies (e.g. mouse, rabbit, chicken, goat), and modifying the histological protocol itself (e.g. various buffers, detergents, serums), and using 3,3′-Diaminobenzidine (DAB) enhancement.

One explanation for weak EGFP expression could be that the eTeNT-EGFP fusion protein is concentrated in the axons terminals as it is actively transported in a small volume. Cell body eTeNT-EGFP may be too dilute to enable that larger signal:background ratio. Alternatively, the lower signal could arise from our emphasis on acute silencing. To avoid potential silencing-induced compensatory mechanisms, we collected tissue at 5 days of doxycycline. This time frame was sufficient to detect functional changes, but could be insufficient for robust histological detection of eTeNT-EGFP. Indeed, the beautiful somatic labeling detected in the original viral vector paper was generated following longer durations of silencing (Kinoshita et al., 2012). Another plausible explanation is that our viral titer was too low for robust EGFP-based labeling. In the Kinoshita et al., (2012), the eTeNT.EGFP and AAV titers were x10^11^ and x10^13^, respectively. With our in-house methods and equipment, we were only able to generate virus of x10^7^ and x10^12^, respectively. However, even with lower cell soma concentration of eTeNT-EGFP, terminal levels were obviously sufficient to functionally alter synaptic transmission.

3) From the contour density analysis in Figure 1H-J, there appear to be differences in the relative bilateral distribution of ipsi and contra cells (if I am interpreting the yellow and blue lines correctly) that indicate systematic differences in the distribution of ipsi versus contra LAPNs as you move caudally. Also, there are differences in lamina distribution in L3 compared to L1 and L2 (Figure 1M-O). It is difficult to understand the functional implications or why the authors carried out this analysis without a bit more information. All of the data are normalized to total, so it is difficult to get a handle on real numbers and variation.

These differences are “real”, in that the variability is relatively modest given that the technique has some inherent variability in pipette localization, volume, etc. We did not provide additional information simply because there wasn’t much more to say that would not be purely speculative and since the literature provides little to go on in terms of L2 and L3 differences in interneuron populations/distributions and any functional implications, at least in our opinion.

We decided to perform these experiments given the unresolvable technical issues we had with detecting the double-infected somata. Using identical methods for the cervical viral injections (e.g. age/sex/strain of rats, injection coordinates and volume), we injected CtB to robustly label the somata thereby allowing us to perform subsequent anatomical analyses.

4) The lack of any sort of identification, either by transcription factor or by transmitter phenotype, makes it difficult to generalize to other locomotor networks. Although glutamatergic and GABAergic axon terminals are identified, the source (whether ipsilateral or contralateral, L1,2,3 or elsewhere) is still unclear. Molecularly-defined excitatory and inhibitory spinal interneurons can migrate some distance from their point of origin, but tend to settle in consistent regions. If laminar distribution is an important clue to their identity (sensory, motor, other), it should be more clearly stated.

We have not provided information/discussion along these lines, again because, in the absence of techniques able to classify interneurons in adult rat spinal cord tissue, it would be purely speculative and perhaps detract from the primary point. The transcription factors that classically delineate the various ventrally-derived “V-class” of spinal interneurons are developmentally regulated. Few are expressed beyond postnatal maturation, thus precluding post hoc V-class categorization in our adult rat tissue through immunohistochemical approaches.

The closest approximation of how our LAPNs fit into the transcriptionally delineated locomotor framework comes from Silvia Arber’s group (Ruder et al., 2016). Using intersectional breeding and viral-based fluorescent tagging, she has shown that V0 (Dbx1-expressing) and V2 (Shox2-expressing) progenitor domains give rise to lumbo-cervical spinal neurons. Contralateral projections predominantly arise from V0-Dbx1 while ipsilateral from V2-Shox2. With AAV-flex injections into the lumbar segments of vGlut^On^ or vGAT^On^ mice, she revealed that the bulk are excitatory in nature with comparatively fewer inhibitory lumbo-cervical projections which are preferentially confined to the ventral horn of the caudal cervical enlargement.

We mapped the LAPN laminar distributions and found that these neurons are embedded within regions where proprioceptive afferent input or descending drive access the spinal locomotor circuitry (Noga et al., 1995). However, this is not really conclusive, and we would prefer to limit discussion in this area.

5) Rhythms within a limb aren't effected so I think it's safe to say that these LAPNs are not rhythm-generating. However, without a better idea of the identity of these neurons, one cannot rule out the possibility that they are sensory interneurons relaying proprioceptive or exteroceptive signals. I think this possibility should be raised in the discussion along with a potential pattern-forming motor function.

We agree that we certainly cannot rule out that these neurons are relaying primarily sensory information or sensory-derived temporal information. We have included that in the current Discussion section and have added emphasis to this possibility in the appropriate paragraph, as follows: “Alternatively, it might be derived principally or entirely from hindlimb afferent input carrying temporal information associated with paw contact…”. We hope this is sufficient.

6) Since different speeds of locomotion are used in different behavioral contexts, it is difficult to separate which of these two features is more important with the current analysis. For example, if bilateral synchrony is observed at faster speeds, then behaviors that are slow would not be affected. It would be worth plotting the phase data as a function of speed in the overground behavioral tasks (e.g., Figure 3), to see if bilateral activity becomes more obvious at faster speeds or if it is observed over the entire speed range. Similarly, it would be good to know the speed range/cycle duration of the other tasks (e.g., treadmill stepping subsection “Silencing LAPNs disrupts interlimb coordination independent from the salient features of locomotion.”) to see how they may overlap. For example, the Arber lab observed no effects on slow treadmill locomotion when they ablated LDPNs, only at fast speeds was a deficit observed. I couldn't find the range of speeds used for treadmill locomotion, but these should be reported.

See comment 1 above and the new Figure 5—figure supplement 2.

7) From Figure 5, it looks like the Dox treated mice are capable of moving faster (cm/s) than controls (panels C-F, H-K, M-P). Could this also explain the increased co-contraction? They are operating in a higher speed regime for more time? Plotting phase against frequency in control and Dox treated animals would also help determine whether this is a context- versus speed-dependent phenomenon.

We would argue that the rats are expressing a larger range of speeds during silencing, rather than being “capable of moving faster” and this may be due to natural tendency for a non-alternating pattern to be easier to perform when moving faster. However, we think the main point is that silencing induced disrupted steps throughout the speed range, which immediately distinguishes it from normal speed-dependent gait changes. We would argue that this concern should be alleviated by our results showing speed-dependent changes, since there is a strong relationship between speed and frequency. These data are found in Figure 4 and Figure 4—figure supplement 1 and Figure 4—figure supplement 2, and in the new Figure 5—figure supplement 2. Also, see comment 1 above. This has been added to the final paragraph of the Results section.

Reviewer #2:Procratsky and colleagues provide an anatomical and functional evaluation of long ascending propriospinal neurons (LAPNs) connecting lumbar hindlimb-related segments and cervical forelimb-related segments. They use an elegant method (developed by Tadashi Isa) to specifically and reversibly silence LAPNs in rat, expecting to uncouple fore/hindlimb coordination. Instead, they observe changes in left-right coupling that occur only during non-exploratory locomotion on high friction surfaces. This study is well-presented and the datasets are comprehensive, and the findings are thought-provoking.In addition to the unexpected (and quite puzzling) primary result, there are other key findings of interest, including the identification of spinal neurons involved in locomotion in a context-dependent manner, and a potential demonstration that disruption of locomotor pattern that does not affect the rhythm. Further, it is exciting to see reversable silencing experiments in an animal model aside from transgenic mice. These results are novel and contribute to our understanding of the spinal locomotor circuit.Major comments:1) For the interpretation of manipulation studies, it is essential to know which neurons are being targeted. The CtB and eTeNT.EGFP histology presented does not directly get at this issue. On one hand, the CtB data provides a thorough description of lumbar LAPNs. However, the overlap with the neurons being manipulated with the eTeNT is missing and, as pointed out by the authors, CtB labels fibers of passage which belong to neurons that are not being silenced. This raises the question of whether the eTeNT.GFP directly overlaps with the CtB-labeled populations, just the numbers are less, or are there specific regions where there are eTeNT.GFP neurons within the more widespread CtB labeling? This is not possible to determine with the results presented in Figure 2. A mapping similar to what was performed for the CtB data would allow the reader to compare and would be helpful to assess exactly what is being manipulated.

See comment 1.

2) I'm not entirely convinced one can conclude that the local commissural projections of the LAPNs are minor from the data presented. According to Figure 1—figure supplement 1, it is about 10% of the commissural LAPNs that do have projections to L1 and L5 (30% of 16% in L1 + 45% of 9% in L5 = ~10%). This is not counting any that may project within segments L2-L4 and even if that's just an additional 5%/segment, that could be 25% which is substantial. The tracer would have needed to be injected into a wider region of the contralateral cord and a low overlap observed to make this conclusion (but if the experiment were to be performed in this way, information regarding ipsilateral LAPNs would have been lost).

We anticipated that a much larger proportion of LAPNs would have extensive projections to L1 and L5, and thus our response to the “scarcity” was to refer to it that way. We agree that this does not rule out these projections as important for function, just that, if they are important functionally it is via fairly modest local outputs compared to what might be expected. We have altered the description of these data to indicate that these are “modest” rather than “sparse”. We have indicated the specificity of the experiment being limited to L1 and L5 (second paragraph of Results section) to ensure that the reader does not come to the wrong conclusion.

12) Following the previous point, there may be significant overlap with the V0 and V2a populations here. It is impossible to know for certain as the overlap of the LAPNs manipulated here with genetic populations cannot be determined and the mouse work does not detail the degree to which local vs LPNs are manipulated. Where the presented locomotor phenotype is similar in some ways to the phenotype seen in V0V and V2a mutants (i.e. left/right synchrony are more prominent and observed at a lower locomotor frequency ), there are distinct differences between the findings here and the mouse studies (trot is lost in the mouse mutants but it is present in the rats, the speed profiles are more compressed in the genetic mutants and that does not seem to be the case here, LAPN silencing effects depend on condition, etc.). Can these experiments be considered as complementary and, if so, does this provide additional insight into the circuitry?

The reviewer raises excellent points concerning the similarities/differences between our silencing approach in rats and the experiments using transgenic mouse models, which are clearly important for investigating how spinal circuits organize locomotor behaviors. Despite the dissimilarities in rodent models and methods, we consider these experiments to be complementary as each affords similar, yet distinct insight into how the spinal cord ultimately secures locomotion.

The genetic-based approach in the mouse affords large-scale, proof-of-concept circuit investigations into how the spinal cord broadly organizes movements. While emphasis is usually placed on the lumbar CPG for delineating roles in locomotion (e.g. fictive locomotion preparations in the elegant experiments of Crone et al., 2008; Crone et al., 2009; Dougherty and Kiehn, 2010; Zhong et al., 2010), the alternation-securing Chx10+ V2a interneurons and Dbx1+ V0v interneurons are actually distributed network throughout the entire spinal neuraxis (Francius et al., 2013) as well as supraspinal centers such as the forebrain (Dbx1+; Causeret et al., 2011) and medullary reticular formation, which in itself is a powerful modulator of locomotor behaviors (Chx10+; Bretzner and Brownstone 2013; Bouvier et al., 2015). Thus, it is somewhat unsurprising that a knockout/manipulation of a broadly distributed network based on its expression of a post (or pre)-mitotic transcription factor leads to a more prominent “all-or-none” phenomena, such as the striking “locked-in” hopping phenotype observed in V0-deleted mice (Talpalar et al., 2013).

Distinct, yet complementary to the large-scale approach described above is our comparatively small-scale, circuit-based approach. These are anatomical-driven investigations of the spinal locomotor circuitry and are designed to acutely and reversibly manipulate select pathways in the otherwise intact and mature nervous system, negating the potential confounding influences of compensatory mechanisms (e.g. compensation following developmental knockouts or plasticity following irreversible ablations) or unintended “off-target” effects (e.g. manipulating a broad and distributed class of neurons which is comprised of multiple subtypes of various functions). This subtler approach affords insight into not just the functional role (e.g. left-right hindlimb alternation), but also the functional importance of discrete pathways in securing locomotion. For example, we find that LAPNs are necessary for alternation during overground stepping, but dispensible for alternation during swimming.

4) It seems that the figure that was uploaded as Supplementary Figure 5 is incorrect. The figure is identical to Figure 5 but figure legend does not match.

We apologize for this mistake. It has been corrected. We are particularly disappointed about this mistake because we believe the data presented speaks to some of the concerns voiced regarding speed and phase.

5) In the Discussion section, although it was clear that the authors expected that forelimb and hindlimb coordination would be disrupted and the lack of that finding is well-described, there is no discussion about why this is not the case. Additionally, the suggestion of LAPNs being "distributors of temporal information" comes up a few times (Discussion section). How that may result in the locomotor changes seen in the data is not obvious and it would help to have an expansion of that idea for clarification.

We agree that this was not included in the discussion at any level. To correct this oversight we have added the following to the first paragraph of the Discussion section. Together, these studies indicate that inter-segmental projecting lumbar pathways are key distributors of temporal information that can be used for maintaining left-right alternating during overground locomotion, and that hindlimb-forelimb coordination is either secured by other means or is less vulnerable to disruption potentially requiring silencing of larger numbers or a wider range of long-propriospinal neurons.

Reviewer #3:[…]We have one substantive concern to raise that is related to whether there is a speed-dependence related to the loss of coordination. This issue stems from data illustrated in Figure 5 where steps in Dox-treated animals are colour-coded to show the steps that are unaffected (red) and those that are affected (blue). It looks like affected steps (blue) are on average occurring at higher speeds than those that are not (red). This suggests that loss of coordination is more likely at higher speeds in Dox-treated animals. Could it be that the differences seen in a different context are purely related to the speed of locomotion achieved in these different locomotor activities? The Dox-treated animals appear to have a greater proportion of strides performed at a fast pace (>90 cm/s) than controls in the volitional locomotion experiment (Figure 5C-P). Perhaps re-analyzing using a sample of steps uniformly distributed across speeds in control and Dox-treated animals could answer whether loss of coordination is speed-dependent or context-dependent.

This question/concern has been addressed in #2. The key point is that disrupted steps occurred at all speeds and, even in the different contexts there is no relationship suggesting that silencing induced higher speeds at which the disrupted steps occurred.

In the same line as above, we did not find details related to the speeds tested for the treadmill experiments. Were the animals tested on the treadmill at multiple speeds (ie. slow, medium, fast)? If so – what speeds were tested and were any differences noted? Similarly, were the average walking speeds different between trials on uncoated and Sylgard-coated surfaces? These details could provide a common factor that explains the apparent context-dependence of the data.

This question/concern has been addressed in comment 2.

18) Finally, while the data clearly demonstrate abnormalities in intra-girdle patterning following inhibition of the LAPNs, the basis for these abnormalities isn't clear. The effects on the hindlimb pattern are particularly perplexing if we assume that since the LAPNs are unlikely to directly interact with the lumbar CPG based upon their sparser connectivity within the lumbar spinal cord. Rather than the LAPNs being necessary for intra-girdle coordination, is it not more plausible that they act by compensating for the proprioceptive confound caused by forward propulsion from the opposite girdle? This could also help to explain why volitional locomotion – where there is likely more variation in speed (and thus more sensory confounds from acceleration and deceleration) – exhibited more gait abnormalities than treadmill walking. While this difference in interpretation might seem trivial, it would help to fit these results within the context of spinal cord injury literature – wherein loss of LAPN connections to the cervical spinal cord by thoracic transection has not (to our knowledge) produced similar forelimb gait abnormalities. In these spinal cord injured animals, the hindlimbs do not produce forward propulsion, and therefore do not have a destabilizing influence on forelimb proprioception/CPG function – thus not necessitating LAPN input to maintain appropriate forelimb alternation. Further discussion or justification for the current interpretation of the results should be provided.

We totally understand viewpoint and hope that the extensive analysis of speed-dependent changes has alleviated these concerns. See comment 2.

[Editors' note: further revisions were suggested prior to acceptance, as described below.]

Summary:The paper has been much improved by new experiments, clarifications, and rewriting. Nevertheless, significant changes are needed before the paper can be published in eLife.Essential revisions:Quantification (alternative changes provided):In response to our comments the authors have provided a viral labeling method that better matches the approach used to silence neurons. While it is still not the same construct, it is a closer approximation. Indeed, this demonstrates that the method used to characterize LAPNs in Figure 1 overestimates the populations labeled. However, the authors have chosen to focus exclusively on ipsilateral populations, when data from Figure 1 clearly illustrate that commissural populations make up a larger fraction of the LAPN population (significant difference, little overlap in values, but downplayed in the main test). The new data are included as 'supplemental', when these are the key experiment to demonstrate which populations are labeled. Since much of the analysis in Figure 1 is not raised in the results and since the authors have already published data describing the populations of LAPNs in the lumbar spinal cord, this figure no longer seems relevant. Instead we suggest one of the following:Alternative 1a) A more detailed analysis of the new viral labeling strategies and their distribution on ipsilateral and contralateral sides of L1-2 to start out as Figure 1. This would provide more space to outline the caveats of the approach to the reader up front.b) Move current Figure 1 into supplemental and provide some context and interpretation to the analysis you have performed and how it differs from their previous study (Reed et al., 2006). The authors say in their response that they do not want to speculate, but at least accurately reporting differences in distribution where they exist (e.g., L3 versus L1-2) is warranted. Particularly since labeling in L3 looks more like their new viral data. Also, please use bold colors instead of a gradient to mark laminae, since this is challenging to discern distinct laminae based on subtle shade differences. For this figure, it would also be more helpful for ease of interpretation, if the contour density analysis was pooled so that contra and ipsi populations are on different sides, not as they are based on dye labeling approaches (where one gets the impression one dye is better at labeling than the other or that contra populations also have ipsi projections or both).Alternative 2a) Remove current Figure 1 altogether and replace with more detailed analysis of the new viral labeling data.

Regarding the new viral labeling strategy, we focused on mapping the ipsilateral LAPNs in an effort to reduce the potential for artefactual labeling of fibers of passage. We believe this gave us a less ambiguous approach to compare CTB-labelled vs dual virus infected LAPNs. This rationale was not provided in the previous response and we apologize for any confusion it may have raised.

Our original intent for the CTB experiments was to determine how LAPNs are embedded within the lumbar spinal gray matter. Our previous anatomical research did provide quantitative data to address this question. Given the rhythmogenic capacity of the lumbar intermediate gray matter, we designed our CTB experiments to address this primary question (in addition to their projection patterns, which we described previously in Reed et al., 2006). To this end, the original CTB data achieve this goal.

We thank the reviewers for their suggestions. We have opted to pursue the Alternative 1b revision. The following changes have been made:

a) The CTB dataset (original Figure 1) has been moved to supplemental (Figure 1—figure supplement 1). Therefore, all CTB related data are now shown in Figure 1—figure supplement 1.

b) In the Figure 1—figure supplement 1, we switched to bold colors instead of a gradient to demarcate the laminae.

c) We appreciate the suggestion for revising the heatmaps and contour plots to aid in interpretation. We have since re-written our Matlab code to pool data generated from both CTB tracers to create a master contour (and heatmap) plot. Per the reviewer suggestion, we now show ipsi- and contralateral LAPNs on different sides (Figure 1—figure supplement 1). We have made the code openly accessible at GitHub (details provided on the Key Resources Table).

d) In terms of context, given the rather crude analysis we performed at the time (Reed et al., 2006) we believe the Fluororuby and current CTB approaches gave largely similar outcomes with respect to a broad characterization of LAPN anatomy. The only apparent differences are the relative lack of deep dorsal horn neurons in the L2 segments in Reed et al., as compared to the current data set, and these differences may likely be due to the overall lower numbers of neurons labeled with the less-efficient FR. We have removed the viral-based ipsilateral only labeling included in the last revision, so no further description or discussion is necessary in our opinion. The CTB labeling presented in supplemental figure 1 provides the needed anatomical description of the LAPN population being targeted for silencing.

Specificity of labeling (alternative changes provided):The bright local labeling presented in Figure 2 is explained in the response to reviews, but not in the main text. The utility of this figure is also not clear, since the new labeling data better serve this purpose. Also, the new data suggest that axons of passage can be labeled by their CTB label, yet there are no L2 neurons labeled by injecting in L1 (Figure 1—figure supplement 1). Some clarification here is required, since the current argument is that spinocerebellar neurons are labeled as axons of passage. The point of this experiment is also not clearly articulated and just adds confusion.We suggest one of the following:Alternative 1:a) Moving Figure 2 to supplemental, with a stronger statement or data supporting the idea that this does not reflect more proximal ectopic labeling. For example, were sections performed in cervical or thoracic levels to rule this out? Is this technically impossible or controlled for in past work? In addition, marking the laminae in which the labeled neurons are located would help link to new viral labeling work.b) Provide a better justification for the experiments and an interpretation of the lack of labeling using local injections.Alternative 2:a) Removing Figure 2, since the new labeling experiments serve this purpose and do not raise as many questions.

We believe there might be some confusion regarding Figure 2. Data shown in the original Figure 2 are not CTB, but instead the histology from the eTeNT.EGFP silenced animals. The utility of these data is validation of the silencing constructs. Data shown confirm eTeNT.EGFP expression at the level of the terminal field and somata during Dox2^On^. Given this confusion, some of the Figure 2 revisions suggested are not applicable.

If the revisions suggested were indeed related to the CTB experiments and not eTeNT.EGFP, we would like to provide the following clarifications:

1) “The utility of this figure is also not clear, since the new labeling data better serve this purpose.”

As stated above, the utility of original Figure 2 was to validate the eTeNT.EGFP silencing construct through histological approaches. Given that we have moved original Figure 1 (CTB dataset) to supplemental per reviewer suggestions above, the original Figure 2 (eTeNT) is now Figure 1. We have updated it to include a schematic illustrating the injection paradigm and experimental design (Figure 1A).

2) “ Also, the new data suggest that axons of passage can be labeled by their CTB label, yet there is no L2 neurons labeled by injecting in L1 (Figure 1—figure supplement 1).”

We acknowledge that axons of passage could be labelled with CTB. However, it is also important to note that the more restricted labeling observed with the dual virus approach could also be by virtue of the technique itself. Given that neurons labelled must be double-infected via retrograde delivery of Cre followed by local delivery of Cre-sensitive AAV, and then undergo cre-recombination for fluorescent tagging, we expected fewer neurons to be labelled from the get-go. Nonetheless, our primary goal was to determine where within the lumbar rhythmogenic core LAPNs are embedded. We believe our CTB data adequately address this question.

While not shown in the supplementary figure, there are numerous L2 neurons labelled following injection at L1. This was not included as we have previously reported these data (Pocratsky et al., 2017, Figure 8 panels M-Q).

(3) “The point of this experiment is also not clearly articulated and just adds confusion.”

We hope we have clarified this with our response.

The original CTB dataset meets the original objective of reconciling where within the rhythmogenic lumbar gray matter LAPNs reside. To this end, the new viral labeling data provided using a separate two-virus system further supports the primary outcome from the CTB data (LAPNs reside within intermediate gray matter, laminae 5-8). Performing additional experiments using a dual virus labeling strategy will not provide significant insight beyond what we already show. As these data were reported for the benefit of reviewers and are not the primary focus of this manuscript (functional study), these data have been removed from the previous Supplemental Figure 1. The new Figure 1—figure supplement 1, revised based on reviewer suggestion related to CTB data, now presents the anatomical story for LAPNs.

Data curation (alternative changes provided):Much of the critical data is currently found in supplemental, which could replace current main figure components that convey the same information. For example, in Figure 3, the running mouse illustrations are beautiful in panels c-e, but they take up a large fraction of the figure and add little beyond what is presented in panel b with respect to differences in coordination. Also, Figure 3 sets up the interlimb coordination differences, then moves to Figure 4 which demonstrates no impact on intralimb effects, then moves back to interlimb differences in Figure 5, requiring another cartoon summary of different gaits. Space could be saved for the inclusion of supplemental data if the figures are re-ordered. We suggest one of the following:Alternative 1:a) Re-order the figures, so that a new Figure 1 deals with the method of labeling and distribution of neurons with bilateral analysis, Figure 2 deals with the lack of impact on rhythm generation, as expected. Then Figure 3 and Figure 4 move into the impact on left-right alternation and limb coordination. In particular, moving data from Figure 5—figure supplement 2 into the main figures is strongly encouraged.Alternative 2:a) Keep figure ordered as they are, but reduce redundant information in figures (e.g., introductory cartoons) to make room for supplemental data that make key arguments (e.g., Figure 5—figure supplement 2). For example, panel 3C could be incorporated into panel 3B and panels 3B and 5A essentially convey the same information.

To reduce redundant information per Alternative 2a, we have removed the schematics illustrating the phenotype (Figure 2) and merged data shown in panels a-c in Figure 3 (intralimb coordination).

Data show in the previous Figure 5—figure supplement 2 is plotted in a manner to illustrate the full range of variability observed (speed vs spatiotemporal parameters for the various locomotor contexts). This format is identical to that shown in Figure 4. In the vein of reducing redundant information per reviewer request, we have elected to move panel ee from Figure 5—figure supplement 2 to Figure 5. Therefore, Figure 5 shows the proportion of disrupted hindlimb steps across the various locomotor contexts as well as the corresponding stepping speed range, mean, and standard deviation. Panel ee was selected in particular as it distills the salient points of the context-related speed data into an easy to interpret graph.

Speed versus context control:The revision still does not rule out the possibility that Dox silencing is simply most obvious in contexts when the animal must move at faster speeds. It seems clear that silencing pushes the animals into a faster range of frequencies and the premature adoption of gaits normally used during very fast locomotion. This would be consistent with the impact of disruptions to spinal interneuronal control circuits in mammals and bolster their argument that LAPNs play similar roles in coordination. For over-ground locomotion (Figure 5—figure supplement 2), altered steps dominate the shift to higher speeds (cm/s). For treadmill locomotion, the mouse does not reach the higher range of speeds observed during over-ground locomotion and instead appears to use a strategy more reliant on longer stride times and lower stride frequencies. So, the lack of altered steps could be because the animal is not pushed to move as fast and adopts a different locomotor strategy (that is still impacted by Dox silencing, but not as much because deficits are always more obvious at faster speeds). The same argument could be made for exploratory mode walking, where the animal is not pushed to move as fast. The coated surface is like over-ground because they can reach faster speeds, while the smooth one is a bit slower (cm/s wise) and so not as impacted. Overall, it seems that deficits evoked by Dox silencing are most obvious in contexts in which the animal needs to move faster, not because they are moving the same way in different contexts however Dox silencing preferentially impacts one context. To better rule out this possibility, we suggest the following:a) Statistical tests of the overall speeds between tasks in addition to the within-tasks analysis of altered versus unaltered that you've already performed. This is presented in Figure 5—figure supplement 2E, but no stats are reported, and it looks like the movements most greatly impacted are ones that have the highest range of frequencies (over-ground and coated, grey bars). If this is significant, the idea that speed is playing a role needs to be considered in the interpretation presented.

The reviewer indicated that the “altered steps dominate the shift to higher speeds.” We disagree. Data shown in Figure 5—figure supplement 1 panel B clearly shows that only ~4% of altered hindlimb steps occur at speeds beyond that observed at control time point (the “Dox expanded range” shown by the shaded pink-red region). Therefore, 96% of silencing-induced altered steps occur at walk-trot speeds observed during control stepping (normal left-right alternation). In addition, in Figure 5E we show that the vast majority of disrupted steps occurred at speeds that were shared across the behavioral contexts assessed, whether there were many steps disrupted (overground and coated) or few (exploratory and smooth). Finally, disrupted steps clearly occurred throughout the speed range (Figure 4). Also, please note that the data provided is raw, representing all the analyzed steps and is thus transparent. In short, we feel that there is no evidence supporting a relationship between disrupted steps and speed and further analytical efforts are not warranted.